# Both partners' negative emotion drives aggression during couples' conflict
Annah G. McCurry ⬛ ✉, Robert C. May & David I. Donaldson ⬛

Researchers examining conflict between intimate partners believe that the experience and expression of emotion drives aggressive behaviour. *Intra*-personally, increases in negative affect make aggression more likely. *Inter*-personally, theoretical models suggest that each individuals' perception of their partners' emotion also influences aggression, potentially creating a Violence Escalation Cycle. Here, using a lab-based aggression task across a primary study (*n* = 104, *number of trials* = 3095) and a replication (*n* = 58, *number of trials* = 3167), we show that both intra- and inter-personal experiences of negative emotion predict reactive aggression within couples, revealing retaliation but not escalation. Critically, analyses of facial affect reveal that prototypic displays of negative emotions have a compounding effect, leading to dramatic changes in aggression depending on whether one, both, or neither partner expressed negative emotion. We propose a mechanism by which temporal delays (i.e., experimentally imposed forced breaks) reduce aggression by decreasing negative emotional arousal and limiting impulsive action. Our results show that both forced breaks and elective breaks (i.e., extra participant-initiated extensions of the forced break time) reduce aggression, providing exciting evidence that interventions focused on preventing impulsive action when people are in a provoked state can reduce aggression within couples.

Intimate Partner Aggression is widespread, with around a third of dating couples at universities engaging in some form of physical aggression during conflict[1]. As a result, researchers and practitioners are increasingly focused on understanding the social and emotional interactions that occur between partners during everyday conflict, and how these predict the escalation towards aggression[2]. Aggression between partners is frequently reactive in nature—that is, it is impulsive[3,4], and is characterised by strong negative emotional arousal[5–7]. Critically, emotional arousal and impulsivity are linked, such that acute negative affect (e.g., an angry state) is associated with failures of self-control. As a personality trait, the tendency to act rashly when upset is called "negative urgency"[3,5–8], and it has been implicated as a central contributor to intimate partner aggression. By contrast, the systematic pattern of abuse sometimes known as 'intimate terrorism' or "coercive controlling violence' is less strongly associated with impulsivity and we therefore use the term 'intimate partner aggression', rather than 'intimate partner violence', to describe our research focus[9].

Within dyads, theoretical accounts such as the General Aggression Model[10], and the I³ model[11], further argue that emotions from both one's self (*intra*-personal) and one's partner (*inter*-personal) influences one's aggressive behaviour. Consequently, if reactive aggression between couples requires negative arousal (influenced by both partners) and impulsivity, we reasoned that it should be possible to reduce behavioural aggression by

experimentally blocking impulsive action while participants are in a provoked state. In principle, impulsive action can be prevented by introducing a 'forced break' period (e.g., a brief experimentally imposed delay) between provocation and the opportunity for aggression. If impulsivity during a provoked state is critical, we would expect a forced break to limit aggression by blocking impulsive action and reducing negative emotion. In essence, assessing the consequences of an experimentally imposed forced break on the dynamics of dyadic conflict provides a strong test of the influence of both negative affect and impulsivity during provocation, allowing us to examine current accounts of Intimate Partner Aggression.

Negative urgency is believed to reflect a failure to appropriately regulate negative emotional arousal. As noted above, negative emotional arousal (and poor emotion regulation ability) has been robustly shown to increase the likelihood of aggressive behaviour within individuals[12–16]. Indeed, three recent meta-analyses all conclude that the inability to regulate negative emotions correlates with reactive aggression[14–16]. Across these studies, emotional arousal has been assessed using a range of methods, including self-report ratings[17,18], heart rate[19], and the analysis of facial expressions[20], where prototypic displays of negative emotion (specifically anger and disgust, the so-called 'moral emotions'[21,22]) are particularly relevant to conflict and aggression. More recently, researchers have suggested that emotional arousal and regulation *between* partners (known as co-regulation) may also

The University of St Andrews, St Mary's Quad, St Andrews, Scotland, UK. ✉e-mail: am650@st-andrews.ac.uk

help to explain aggression in couples[16,20,23,24], though robust experimental evidence is lacking due to a paucity of research using face-to-face dyadic designs (i.e., where two people engage in an aggression task together). Research involving couples is clearly essential to fully understand the impact of dyadic affective processes on conflict, particularly the impact of one partner's emotional expression on the other partner's behaviour. Consequently, in the present study we examine behaviour using real couples engaging in competitive interactions face-to-face. One advantage of employing a dyadic aggression task is that this approach can be combined with real-time estimates of emotional arousal, such as monitoring facial expressions, allowing the temporal dynamics of conflict to be assessed.

While experimental studies of dyadic conflict are rare, studies using self-report measures routinely suggest that most couples' aggression found in community samples (i.e., not clinical, forensic, or offender populations) is bidirectional, meaning that both partners act aggressively during conflict,[25–27]. Given the bidirectionality of most couples' conflict, the Violence Escalation Cycle[28] predicts that aggression within dyads will involve retaliation that escalates over the course of a conflict. Specifically, the Violence Escalation Cycle posits that when a person aggresses against their partner, the partner will perceive an injustice and retaliate with slightly more aggression, wherein the original aggressor will perceive an injustice and retaliate as well, resulting in a cyclic pattern of escalation. Despite prominent theories[10,28] (and some research)[28] attesting to the escalatory nature of couples conflict, to date, there is little experimental evidence pertaining to retaliation and escalation, at least in part due to the paucity of experimental, dyadic research noted above.

In response, the current study asks three questions relating to the role of dyadic affective and behavioural processes in reactive aggression between romantic partners: First, does the introduction of a forced break (i.e., a brief experimentally manipulated delay ranging from five to 15 s) between provocation and the opportunity for aggression prevent impulsive action and reduce negative emotions, thereby decreasing aggression? If the answer to this first question is "yes", then do longer breaks produce greater reductions? Second, do measures of emotional expression from both partners (assessed via automated machine learning assisted affect coding of videos taken during dyadic interactions) provide evidence of co-regulatory interaction, modulating an aggressor's actions? Third, does aggressive behaviour between romantic partners exhibit escalating retaliation over the time course of a conflict when couples are engaged in an on-going competitive interaction? Overall, therefore, our aim is to assess the ways in which emotions experienced by both partners during a conflict are predictive of aggressive behaviour.

To answer these questions we adapted a common aggression paradigm (the Competitive Reaction Time Task; CRTT; similar to the Taylor Aggression Paradigm (TAP))[29] to allow intimate partners (adults in romantic relationships; chosen for our focus on intimate partner conflict) to compete face-to-face in a multi-round reaction time 'game' (see Fig. 1a). To provide an ethically acceptable opportunity for participants to exhibit aggressive behaviour, the paradigm requires the winner of each round to select the volume of a noxious noise (a 'sound blast') to send to their losing partner's headphones (see Fig. 1b). In the 'immediate response' control condition (which functions as our baseline), the winner is allowed to select a blast level immediately after winning, whereas in the 'forced break' experimental conditions, the winner is blocked from selecting a blast level for 5, 10, or 15 s after winning. We include multiple break lengths to potentially identify the most effective delay (i.e., the largest decrease in aggression with the lowest time cost).

As well as using trial-to-trial changes in blast level as an empirical measure of aggressive behaviour, all participants completed a self-report measure of trait aggression (BPAQ-SF: the Buss and Perry Aggression Questionnaire, Short Form). In addition, we used OpenFace 2.0[30] (a machine learning program for automated Facial Action Coding[31]) to identify prototypic displays of emotion during the blast initiation response time window (blast initiation ±1 s; see Fig. 1b), which allows us to assess the impact of each participant's emotional experience on behaviour (see Fig. 1c). The differences between positive and negative emotions expressed during the experiment are visible in composite images (created with

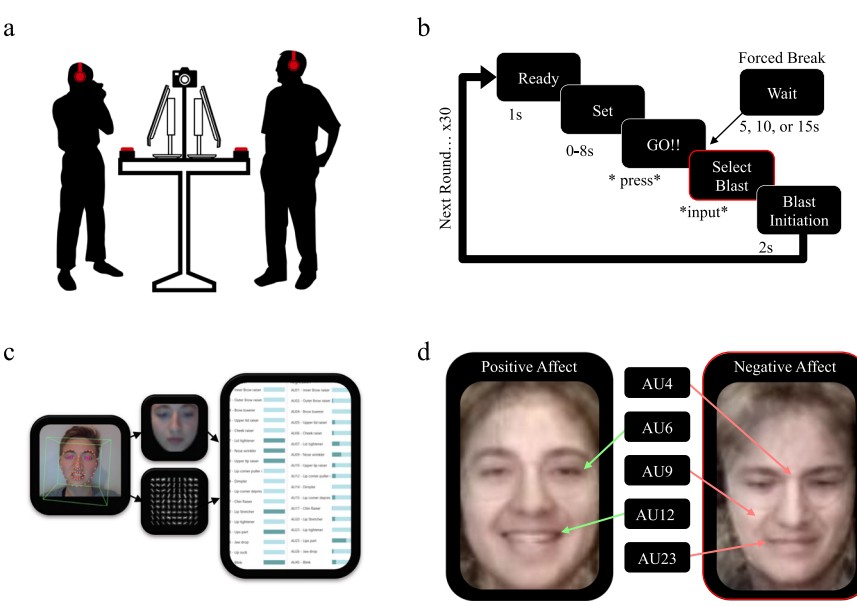

**Fig. 1 | Experimental Set Up and Affect Coding. a** Volunteers (romantic couples) participated together, face-to-face. Each participant had a monitor, a big red button, a keyboard, and a pair of headphones. To record facial expressions an Insta360 camera was positioned centrally, below the eyeline, allowing participants to see each other unobstructed. **b** Each round began with the prompt "Ready" (in white text on a black screen), followed by "Set" exactly one second later, and finally "GO!!" after 0 to 8 s (varied randomly to increase task attention). Screens then revealed who had pushed their button first and prompted the winner to select a blast level (either immediately or delayed by a 5, 10, or 15 s forced break, depending on the experimental condition). Once a blast level was selected by the winner, the noise was delivered at the chosen volume to the loser's headphones for 2 s. Once the blast was over, the next round began immediately. Participants played 30 rounds. **c** Video recordings were analyzed frame-by-frame using a machine learning program designed to automate annotation with the Facial Action Coding System, allowing prototypic displays of emotion to be identified. **d** Composite facial images illustrate differences in expression for prototypic displays of positive and negative emotion. Images were generated by averaging together stills exhibiting the characteristic Action Units (AUs) associated with each effect. AUs illustrated are 6 (cheek raiser) and 12 (lip corner puller) for positive effect; 4 (brow lowerer [sic]), 9 (nose wrinkle) and 23 (lip tightener) for negative effect.

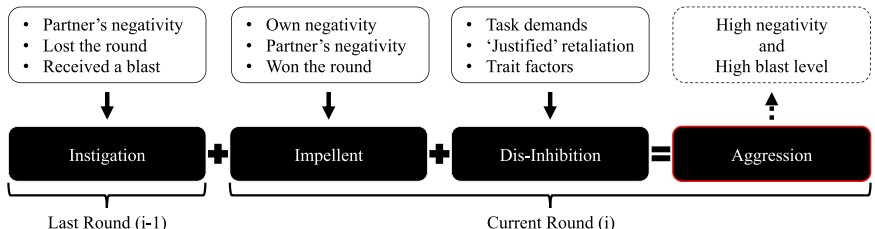

**Fig. 2 | Illustration of the aggression eliciting effects expected from our face-to-face CRTT.** According to the I[3] Model and Perfect Storm theory. In the last round (i-1), a player loses, is faced with their partner's negative emotions, and is blasted with a loud, unpleasant sound. In the next round they win (i): having been instigated by their last loss, they are impelled toward aggression by their own negative emotions,

their partner's negative emotions, and the fact they won the round. Finally, the task demands of the paradigm, along with a sense of justified retaliation for previous blasts and latent trait factors, act to disinhibit participants. Theoretical models predict that this environment should produce aggression, resulting in high negative emotional expression and high blast level selections.

PsychoMorph[32]; see Fig. 1d). Importantly, because we measured facial expressions continuously throughout the task, we were able to examine how behaviour changed as a function of the emotions experienced by each partner. Further, and whenever possible, we use trial-level data in our analysis to maximize power. The use of trial-level data enables us to examine the rich relationship between each partner's blast levels and affect over time, and potentially solve a historic problem in the field (i.e., how to study aggression in lab settings when the level of aggressive behaviour that can be ethically elicited is minimal). A solution here is possible because trial-level data enables us to extract instances of aggression within a context of overall non-aggressiveness (i.e., an ethically permissible lab-based aggression paradigm). From a theoretical perspective, therefore, our paradigm includes the features that are expected to produce behavioural aggression according to the I[3] model and Perfect Storm Theory[2] (instigation/provocation, an impellent, and dis-inhibition, see Fig. 2). Finally, in addition to our primary study, we also present a replication that provides additional support for the initial findings and includes qualitative reports of the impact of forced breaks on our participants, illuminating the mechanism behind forced breaks.

## Methods
### Primary study
**Participants**. A total of 104 participants (60 women, 3 non-binary participants; assessed using the item "What is your gender?" in the post-experiment questionnaire) were recruited from a participant pool at the University of St Andrews in Scotland. Participants were romantic couples (of any orientation) who participated together, and each partner was given £12.50 in compensation. Participants' mean age was 21.1 years (SD = 3.3) with a minimum of 18 and a maximum of 39. Most participants were full time students (72.8%) and white (70.3%). Most participants were not cohabiting (66.1%), and the average duration of a relationship was roughly eight months. This study was approved by the School of Psychology and Neuroscience Ethics Committee in December 2022 (approval code PS16636) before data collection commenced. Consent from participants was obtained in person with physical consent forms; procedure described below.

**Design**. We used a between-groups design. Couples were assigned randomly upon arrival (and without their knowledge) either to the immediate response ($n = 23$) or one of the forced break conditions (5 s: $n = 32$; 10 s: $n = 23$; and 15 s: $n = 26$). The study was not pre-registered.

### Materials
**Trait Aggression**. We assessed trait aggression using the short form of the Buss and Perry Aggression Questionnaire (BPAQ)[33]. The BPAQ-SF is a self-report measure that asks participants to rate statements (e.g., "I have threatened people I know", "I have trouble controlling my temper", etc.), judging how well the statement describes them on a five-point scale from "very unlike me" to "very like me". Higher scores indicate higher trait aggression.

**Acute Aggressive Behaviour**. Acute aggressive behaviour was measured using a modified version of the Competitive Reaction Time Task (CRTT)[34]. In the classic CRTT, a participant plays a competitive reaction time game against a fictitious virtual opponent, and the (rigged) winner of each round sends a noxious white noise "sound blast" to the loser (where the winner chooses the volume, duration, or both). Extending Anderson and colleagues'[28] modification where two real participants unknown to each other played virtually from separate rooms, we brought romantic partners into a single room, together, and had them play face-to-face[28]. According to the recommendations of Elson and colleagues[35], participants could modify only the loudness of the noise blasts (and not the duration) on an eight point scale ranging from 75db to 110db, played for 2 s (with an instantaneous rise time)[35].

Our Face-To-Face CRTT (FTF-CRTT) was partially rigged. When participants' reaction times were within 100 ms of each other, a winner was selected at random. When one participant's reaction time was more than 100 ms later than their partner, the faster participant won the round. This was designed to avoid suspicion of rigging (pilot testing showed that participants could auditorily distinguish whose button press reaction time was faster when there was more than a 100 ms gap) while keeping the win-loss rate of each participant close to 0.5 (so that all players had roughly equal opportunity to aggress). Higher scores (i.e., where the winner chooses a louder blast to play to the loser of the round) are indicative of more aggression. For the purpose of comparing trial level blasts over time, and to allow graphical representations of the interaction between couples, blast levels during win streaks (i.e., where a participant wins two or more rounds in a row) were compressed and the mean of the streak represents the blast level for the given time interval (and the mean of negative expression represents negativity expressed across the win streak).

**FACS and OpenFace**. FTF-CRTT gameplay was recorded using an Insta360 camera placed in-between participants below face height (so couples could see each other unobstructed). Videos were then exported twice as standard mp4 files (each focusing on one partner). Action units of the Facial Action Coding System (FACS)[36,37] were extracted automatically by OpenFace[38], a machine learning program designed to automate FACS scoring. Once action units were extracted, the scoring for specific emotions was coded using R[39] according to the FACS and Emotion FACS (EMFACS) researcher manuals (not publicly available). OpenFace provides both binary read outs of action units (i.e., present or absent) and intensity readouts on a scale from zero to five (based on the FACS intensity scoring system). To maintain homogeneity with the FACS, we used the intensity readouts in our analysis. Higher intensity scores indicate stronger prototypic emotion expression, which is assumed to index experienced emotional arousal. Prototypic emotional displays were coded according to the core combinations introduced by Ekman and Friesen[36,37], modified to fit EMFACS coding rules and use with OpenFace (see Table SI1 of the Supplementary Information for details). Negative emotion included prototypic displays of anger and disgust, whilst positive emotion included happiness. FACS scoring was conducted

by AGM, who is certified to use the FACS and the EMFACS through the Paul Ekman Group.

Having identified the intensity of negative or positive emotion during the blast initiation response time window (blast initiation ± 1 s) for each player, throughout the task, we used selective averaging procedures to examine blast level selections as a function of emotion. High levels of negative affect were operationally defined as trials where negative expressive power was greater than one standard deviation above the mean (with all other trials categorised as low). High levels of positive affect was defined as trials where positive expressive power was greater than one standard deviation above the mean (with all other trials categorised as low) and negative affect was low. To examine compounding effects, blast level selections in the 'immediate response' condition were categorised according to whether each member of each couple exhibited a high level of negative affect (defined as above, examined independently in each player).

**Procedure.** The study was advertised using the University of St Andrews' internal participant recruitment and memo systems as a study of competitive reaction time (omitting our focus on aggression). Interested participants completed an online pre-screen to assess inclusion criteria (i.e., over the age of majority in Scotland, currently in a romantic relationship). Following form submission, an experimental session was scheduled and both partners came to the lab together. Upon arrival, participants read information sheets and signed consent forms. After being given an opportunity to ask questions, participants were shown a demonstration of the game, where they were able to hear the lowest, medium, and highest blast levels (1, 4, and 8, played at 75 db, 90 db, and 110 db, respectively). Participants were then prompted to put on their headphones and stand in front of their monitors. Each participants had a set of headphones, a large arcade button, and a keyboard (see Fig. 1a). The participant stations were set up such that participants stood facing each other, with no visual obstructions. Our headphones were not noise isolating, and partners could communicate verbally. The 360° camera was positioned between participants, just above the level of the monitors and below the eyeline. Video was recorded in 5.7k at 30 fps.

Once participants were ready to begin, our custom FTF-CRTT interface (coded in Python[40]) appeared on participants' screens. They were prompted to enter their names and begin the game. Each round started with a black screen displaying "Ready" in white text. One second later, "Set" appeared, and then 0–8 s later (time length was randomized to increase task attention), "GO!!" appeared, signalling the participants to hit their large arcade buttons as quickly as possible. The names of the winner and loser of the round were displayed. The winner was then prompted to enter their selected blast level on their keyboard and the blast was sent to the loser's headphones. The next round would begin immediately once the blast ended, and the game continued for 30 rounds. In the control "immediate response" condition (which functions as our baseline), participants were able to select the blast level straight away, whereas in the experimental "forced break" conditions, participants were unable to select a blast level for 5, 10, or 15 s after the winner was announced. We used a between subject design and multiple forced break lengths to assess whether the impact of forced breaks varied by their length.

Once all 30 rounds were complete, participants filled in a post-experimental questionnaire containing the BPAQ-SF and custom measures of self- and other- emotion and aggression. We also prompted participants to (optionally) indicate how they felt about their experience overall in a free text answer box as part of the debrief. Participants were then debriefed and told of the true intentions of the study (to examine aggression). All participants were given the opportunity to ask questions about the study and directed to support resources that would be helpful in case of distress. We then recorded their email addresses so that we could deliver compensation, thanked them, and released them.

We did not show participants the blast levels chosen by their partner, but we also did not prevent them from peeking at their partner's keyboard or talking to each other about the discrete numbers selected. Because the loser received whichever blast level the winner chose, we had no reason to try to hide this from the loser, and no practical way to prevent participants from discussing it. In fact, we chose to use non-noise isolating headphones specifically so that partners could talk to each other during the game, to increase ecological validity, and so that the winner could hear (from sound leaked from the loser's headphones) that they were really in control of the blast level.

### Follow-on study

**Participants.** A total of 58 participants (29 women, 1 non-binary participant; assessed by the item "what is your gender?" in the post-experiment questionnaire) were recruited from a participant pool at the University of St Andrews in Scotland. Participants were romantic couples (of any sexual orientation) who participated together, and each partner was given £12.50 in compensation. Participants' mean age was 21.9 years (SD = 7.4) with a minimum of 18 and a maximum of 62. Most participants were full time students (80.0%) and white (83.3%). Most participants were not cohabiting (70.0%), and the average duration of a relationship was roughly fifteen months. This study was approved by the School of Psychology and Neuroscience Ethics Committee (as an amendment) in January 2024 (approval code PS16636) before data collection commenced. Consent from participants was obtained in person with physical consent forms; procedure described below.

**Design.** The follow-on study employed a complex design including both within and between participant manipulations. Couples were assigned randomly upon arrival (and without their knowledge) to one of two conditions that either started with the immediate response task (n = 28, 14 couples), or started with the forced break task (n = 30, 15 couples). In each case a second task was performed, to allow a within participants examination of the effects of task instruction (to be reported separately). Examining behaviour from the first task performed by each group therefore provides a direct replication of the between participants comparison of immediate response and forced break conditions in the primary study. The task and procedures (e.g., instructions, number of rounds, etc.) were matched to the original study, with the exception that in the follow up study we only included a 10 s break condition (as opposed to 5 s, 10 s, and 15 s as in the original study). Here, for the sake of brevity, we only include data from the follow-on study that provides a replication of the original experiment. For the replication data we report all measures and experimental conditions in the present article and the associated reporting summary. The study was not pre-registered.

**Materials.** All materials used in the primary study were used in the follow-on study. In addition, the follow-on study included two additional measures; a test of emotional facial expression recognition ability (used as a filler in between the first and second experimental tasks, data to be reported separately), and several open-ended qualitative questions (presented after the second experimental task was completed, as described below). Given the order of testing we can be confident that neither of these additional measures can have influenced the replication.

Qualitative Questions. In a survey completed at the end of the experimental session, we asked participants to rate three relationship quality factors on a Likert scale from one (low) to seven (high), as in previous research[5]. Further, participants were provided with an optional free-text answer box where they were asked to describe their thoughts and feelings during the game and the forced breaks.

**Procedure.** Procedure in our follow-on study is identical to the primary study with minor modifications. Specifically, participants completed two games (30 rounds each) of the CRTT, separated by a filler task. Following completion of the first game (as described above), participants were each given a laptop on which to complete the filler task (which took

**Table 1 | Summary statistics**

| Variable/comparison | Mean | SD |
|---|---|---|
| **Fig. 3** | | |
| BPAQ-SF | 13.86 | 9.97 |
| Blast Level | 3.72 | 2.31 |
| Positive Affect | 0.83 | 0.77 |
| Negative Affect | 0.12 | 0.21 |
| **Fig. 4** | | |
| *Mean Blast in Immediate Response* | | |
| High Negative | 5.58 | 2.56 |
| Low Negative | 3.59 | 2.15 |
| High Positive | 4.93 | 2.33 |
| Mean Blast in Forced Break | | |
| High Negative | 3.69 | 2.23 |
| Low Negative | 3.63 | 2.29 |
| High Positive | 4.65 | 2.54 |
| *ResponseNegative Affect Level in Immediate Response* | | |
| Winner revealed | 0.29 | 0.17 |
| Blast Selection | 0.32 | 0.41 |
| *Negative Affect Level in Forced Break* | | |
| Winner revealed | 0.48 | 0.28 |
| Blast Selection | 0.15 | 0.22 |
| **Fig. 5** | | |
| Neither | 3.36 | 1.99 |
| Loser | 4.33 | 2.45 |
| Winner | 5.14 | 2.62 |
| Both | 6.24 | 2.34 |
| **Fig. 6** | | |
| Proportion Match | 0.15 | 0.16 |
| Proportion De-escalate | 0.39 | 0.25 |
| Proportion Escalate | 0.46 | 0.25 |
| Mean Escalation | 0.07 | 1.04 |

approximately seven minutes). Once both participants completed the filler task they were brought back over to the CRTT station to play another 30 rounds. Once the second game finished, participants completed the post-experimental questionnaire, and were then debriefed and excused in the same manner as the primary study.

## Statistical analysis (both studies)

**Null hypothesis significant testing statistics.** All analysis was performed in R using two-tailed analysis. All regressions were performed using the Linear Least Squares method. All other comparison of means testing used Wilcoxon Rank Sum tests with a Continuity Correction because Shapiro-Wilk Tests for Normality revealed that all data was significantly non-normal (see Table SI2 of the Supplemental Information); equal variance was assumed. Cronbach's Alpha[41,42] indicated that internal consistency was "robust" or better[43] for each measure (FTF-CRTT, 30 items, $\alpha = 0.925$; negativity, 30 items, $\alpha = 0.866$; BPAQ-SF, 12 items, $\alpha = 0.835$). The effect size, $r$, was calculated from the Wilcoxon tests using the wilcox_effsize function of the rstatix package (version 0.7.2). Standard summary statistics are available in Table 1 and full statistical information is available in Table 2. Where possible, analysis was conducted with trial-level data to extract as much information from the data recorded as possible and avoid biasing results with artificially normal (averaged) data with deceptively low variance[44]. Finally, Multi-Level Modelling and Actor-Partner Interdependency Modelling were

conducted, using the gls function from the nlme package (version 3.1–164) and the lmer function from the lme4 package (version 1.1–35.3), respectively. Normality of residuals for all models was assumed.

**Bayesian statistics.** All analysis was performed in R using the *BayesFactor* package (version 0.9.12–4.6). For comparison of means testing, the *ttestBF* function was used with Jeffreys prior applied to the variance of the normal prior sample (i.e., a minimally informative prior) and a Cauchy prior applied to the effect estimate[45,46]. For correlation analysis, the *regressionBF* function was used with the same arrangement of priors. All grey-shaded regions shown in plots represent standard errors. All boxplot elements represent their defaults in ggplot2 (i.e., centre line, median; upper and lower box boundaries, Inter Quartile Range (IQR); upper and lower whiskers, smallest or largest value within 1.5× the corresponding IQR; dots, outliers).

## Reporting summary

Further information on research design is available in the Nature Portfolio Reporting Summary linked to this article.

## Results

### Primary study

As in previous studies[47] the BPAQ-SF (*max score* = 60[cf. 33]) shows that a convenience sample of mostly undergraduate students scores lower ($m = 13.86$, sd = 9.97) on trait aggression than the general population. Similarly, as in previous research using the CRTT[48,49], the standard experimental measure of aggression (all blast levels, averaged across trials for each participant) did not significantly correlate with participants' self-reported trait aggression (see Fig. 3a: $t(105) = 1.05$, $p = 0.29$, $r = 0.10$, 95% CI = −0.09–0.29, bf = 0.40). Poor correlations between the BPAQ and standard versions of the CRTT have been widely reported and discussed in the field[34,35,50] and we will return to this in the Discussion.

However, our facial affect data (see Fig. 3b) potentially explains the inconsistent CRTT findings in the present literature: prototypic displays of positive affect (happiness[20,21,51,52]) were far more frequent and intense than prototypic displays of negative affect (anger and disgust, both associated with conflict behaviour[20,21,51,52]; *mean positive affect* = 0.83, SD = 0.77; mean *negative affect* = 0.12, SD = 0.21; difference: $w = 8e + 6$, $p > 2e\text{-}16$, $r = 0.50$, 95% CI = 0.61–0.68, bf = 7e + 484). In other words, participants appear to find the CRTT task to be more fun than upsetting, a pattern also reflected in debrief feedback from participants. High negative emotion was present sufficiently often, however, (16% of trials) that its impact on aggression could be examined using selective averaging procedures, as shown below.

**Negative emotion drives aggression and forced breaks reduce it.** Using the facial affect data, we isolated trials associated with high levels of positive or negative expressions of emotion, operationally defined as trials where positive or negative expressive power (mean intensity) was greater than one standard deviation above the mean during the blast initiation response time window (blast initiation ±1 s). Within the "immediate response" condition, participants used significantly more "highly" aggressive behaviour (blast levels 6, 7, and 8) on high negative affect trials compared to high positive affect trials (illustrated as a density plot in Fig. 4a: $w = 2871$, $p = 7e\text{-}3$, $r = 0.23$, 95% CI = 4e-5–2.0, bf = 5.4). Among trials with high levels of negative affect, we observed significantly lower blast levels in all the "forced break" conditions, compared to the immediate response condition (illustrated in Fig. 4b: $w = 8002$, $p = 5e\text{-}4$, $r = 0.34$, 95% CI = 1.0–3.0, bf = 2e + 5). Pairwise comparisons across our three forced break conditions revealed no significant differences in blast levels as a function of delay length (a full breakdown of all forced break conditions is provided in Tables SI3, 4, and 5, and Figure SI1 of the Supplemental Information). Consequently, for simplicity our analysis presented in the main text combines all forced break conditions together. Consistent with the reduction in aggressive responding, analysis shows a positive association between negative expressive intensity during the

## Table 2 | Full Statistics

| Sample | Test | Panel | n | k | Effect Size | P-value | Bayes Factor | 95% CI ↓ | 95% CI ↑ | df | Stat |
|---|---|---|---|---|---|---|---|---|---|---|---|
| **Fig. 3** | | | | | | | | | | | |
| Full Sample | Least Squares Regression | a | 104 | NA | 0.10 | 0.29 | 0.40 | −0.09 | 0.29 | 105 | 1.1 |
| Full Sample | Wilcoxon Signed-Rank Test | b | 104 | 3095 | 0.50 | **<2e-16** | 7e + 484 | 0.61 | 0.68 | NA | 8e + 6 |
| **Fig. 4** | | | | | | | | | | | |
| Winning Players | Wilcoxon Signed-Rank Test | a | 23 | 135 | 0.23 | **7e-3** | 5.36 | 4.e-5 | 2.00 | NA | 2871 |
| | | b | 100 | 119 | 0.335 | **5e-4** | 2e + 5 | 1.00 | 3.00 | NA | 8002 |
| | | c | 101 | 1082 | 0.03 | 0.60 | 0.19 | −9e-1 | 8e-7 | NA | 5180 |
| | | d | 104 | 1322 | 1e-3 | 0.93 | 0.08 | −1e-5 | 6e-5 | NA | 1e + 5 |
| | | e | 61 | 168 | 0.64 | **<2e-16** | 6e + 24 | 0.28 | 0.35 | NA | 24519 |
| | | f | 76 | 202 | 0.18 | **3e-7** | 7e + 29 | −0.37 | −0.17 | NA | 1274 |
| Winners with high negativity | Least Squares Regression | g | 81 | 228 | -0.18 | **5e-13** | 9e + 9 | −0.23 | −0.13 | 1548 | −4.2 |
| | | h | 81 | 228 | -0.29 | **9e-6** | 1802 | −0.40 | −0.17 | 226 | −4.5 |
| **Fig. 5** | | | | | | | | | | | |
| Immediate Response Condition | Wilcoxon Signed-Rank Test | Neither, Loser | 23 | 275 | 0.17 | **0.01** | 21.53 | −1.99 | −3e-5 | NA | 5408 |
| | | Neither, Winner | 23 | 251 | 0.25 | **5e-5** | 1e + 4 | −2.99 | −1.00 | NA | 2736 |
| | | Neither, Both | 23 | 237 | 0.34 | **2e-8** | 5e-8 | −4.00 | −2.00 | NA | 1119 |
| | | Loser, Winner | 20 | 110 | 0.15 | 0.11 | 0.69 | −2.00 | 7e-5 | NA | 1181 |
| | | Loser, Both | 21 | 96 | 0.35 | **6e-4** | 47.13 | −3.00 | −0.99 | NA | 548.5 |
| | | Winner, Both | 21 | 72 | 0.24 | **0.047** | 1.01 | −2.00 | 2.57 | NA | 455 |
| **Fig. 6** | | | | | | | | | | | |
| Winning Players | Least Squares Regression | a | 104 | 1550 | 0.03 | 0.27 | 0.10 | −0.03 | 0.12 | 1548 | 1.1 |
| Full Sample | Least Squares Regression | b | 108 | NA | 0.84 | **<2e-16** | 5e + 14 | 0.78 | 0.91 | 56 | 12.6 |
| | | c | 25 | 360 | 0.44 | **<2e-16** | 7e + 29 | 0.38 | 0.49 | 1548 | 13.3 |
| | | i | 104 | NA | −0.87 | **<2e-16** | 6e + 13 | −0.92 | −0.78 | 50 | −12.5 |

Bolded p-values indicate statistically significant results at the 0.05 level.

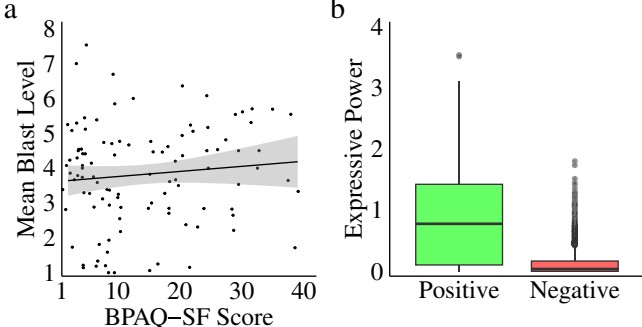

a
b

**Fig. 3 | Trait aggression and emotional expression. a** Behavioural performance data revealed that our self-report measure of trait aggression (the BPAQ-SF) did not correlate with our in-task measure of aggression (mean blast level), consistent with previous research showing mixed results regarding the relationship between the BPAQ and the Competitive Reaction Time Task (CRTT). **b** Our assessment of emotional expressions during the blast initiation period (blast initiation ±1 s) explains the lack of relationship between BPAQ and aggression measures seen here and in previous studies—participants displayed significantly more positive affect (green) than negative affect (red) during task performance. *Defining Plot Elements.* Grey-shaded regions in regression plots represent standard errors. Boxplot features are as follows: centre line, median; upper and lower box boundaries, Inter Quartile Range (IQR); upper and lower whiskers, smallest or largest value within 1.5x the corresponding IQR; dots, outliers.

blast initiation response window (blast initiation ±1 s) and blast level selections in the immediate response condition ($t(345) = 6.6$, $p = 2e-10$, $r = 0.33$, 95% CI = 0.23–0.42, bf = 5e + 7), but not in the forced break condition ($t(1201) = 0.5$, $p = 0.61$, $r = 0.01$, 95% CI = −0.04–0.07, bf = 0.07).

To assess whether forced breaks influenced behaviour more broadly, we also examined the distribution of blasts when positive emotions were expressed. We found no evidence of a reliable difference between the immediate response and forced break conditions on trials with high levels of positive affect, demonstrating the selective nature of the delay effect (Fig. 4c: $w = 5180$, $p = 0.60$, $r = 0.03$, 95% CI = −9e-1 − 8e-7, bf = 0.19). Similarly, we found no evidence that forced breaks elicited a change in behaviour on trials with low levels of negative emotion, where lower blast levels dominated (Fig. 4d: $w = 144438$, $p = 0.93$, $r = 1e-3$, 95% CI = −2e-5–6e-5, bf = 0.08). Finally, we also examined whether there was a reduction in the intensity of the winner's expressed negative affect in the forced break condition between the time the winner was announced (Time 1) and their blast selection after the forced break (Time 2), testing whether the reduction in aggressive responding (illustrated in Fig. 4b) was actually associated with reduced negative affect. Analysis confirmed a significant reduction in negative emotion over time in the forced break condition (Fig. 4e: $w = 24519$, $p > 2e-16$, $r = 0.64$, 95% CI = 0.28–0.35, $bf = 6e + 24$), corresponding to the lower blast level selections observed following a forced break. Moreover, there is a significant difference in negativity change between the forced break and immediate response condition (where the

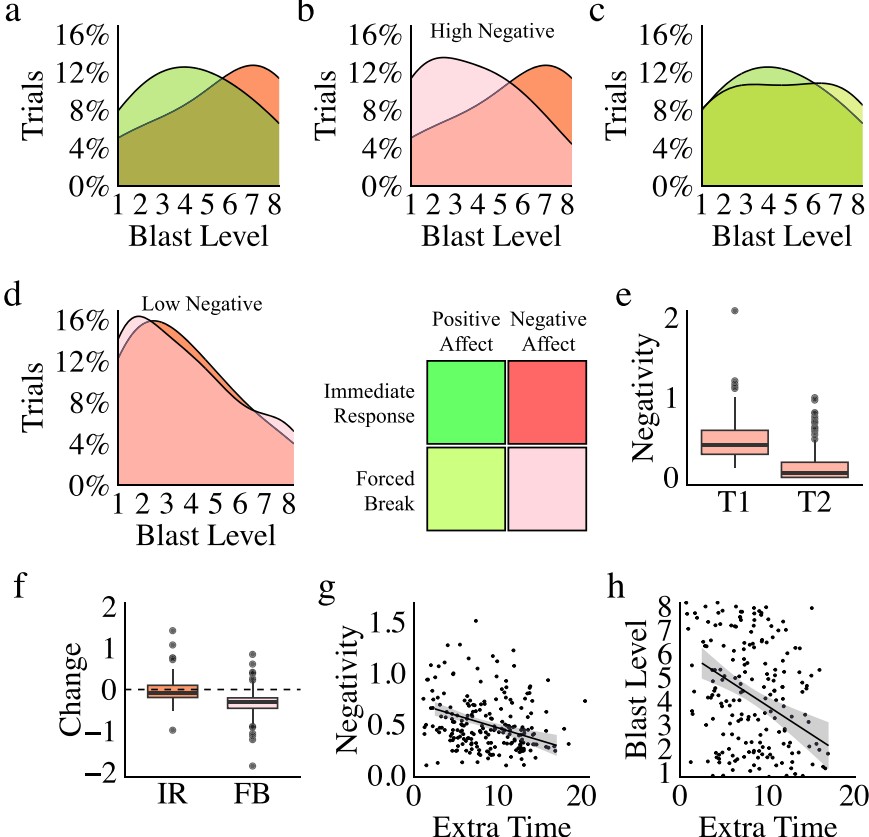

**Fig. 4 | Identifying aggressive behaviour. a** Using facial affect data we isolated responses with high positive or negative emotional expressive power. We then examined aggressive behaviour (the proportion of responses at each blast level) as a function of emotion when participants were able to act immediately, illustrated as density plots. The distribution of responses reveals significantly greater use of high blast levels when *high negative affect* was present (compared to high positive affect without high negative affect). **b** Further examination of behaviour associated with *high negative affect* revealed significantly lower use of high blast levels when responses were made after a forced break (compared to immediate responses). **c** By contrast, examination of behaviour associated with *high positive affect* revealed no effect of a forced break (compared to immediate responses) and no skew towards high blasts. **d** Similarly, examination of behaviour associated with *low negative affect* also revealed no effect of a forced break (compared to immediate responses), reflecting greater use of low blasts. **e** Given that we observed a reduction in aggressive responding in the forced break condition when participants expressed negative affect, we reasoned that negative expression should have been greater when the winner was announced (Time 1), compared to after the break, when the winner selected a blast level (Time 2). As predicted, box plots show a significant reduction in mean negative emotional expressive power over time in the forced break – the reduction in highly aggressive behaviour seen following forced breaks was associated with a reduction in negative affect. **f** Further, the difference in negativity change between the immediate response and the forced break condition is also significant, strengthening the claim that forced breaks produced the observed reduction in aggression. **g** When players elected to wait even longer than their forced break time, the extra time they waited negatively correlated with negative expressive intensity. **h** Extra time waited after the forced break period is over also negatively correlated with aggression. Together, these data suggest that taking a break can be effective whether it is forced or voluntary. *Defining Plot Elements.* Grey shaded regions in regression plots represent standard errors. Boxplot features are as follows: centre line, median; upper and lower box boundaries, Inter Quartile Range (IQR); upper and lower whiskers, smallest or largest value within 1.5x the corresponding IQR; dots, outliers.

T1–T2 difference in the Immediate Response condition comprises the time taken to input a blast level once the winner was announced; Fig. 4f: $w = 1274$, $p = 3e\text{-}7$, $r = 0.18$, 95% CI = −0.37 to −0.17, $bf = 7e + 29$). As well as showing that high levels of negative emotion drive aggressive behaviour within couples, these data confirm that the introduction of a forced break between provocation and the opportunity for aggression mitigates negative affect and results in a significant reduction in aggression.

Further to the above analysis, our data also allows us to examine the impact of couples who, of their own accord, elected to wait longer before choosing a blast level. Our experimental paradigm enforced a minimum break length by blocking blast selection entries for a set period, but participants were not forced to respond immediately after their break was over. As a result, we were able to calculate the amount of extra time participants elected to take after their forced break was over. Analysis of these extra (additional) breaks reveals that voluntarily electing to wait longer than required further reduced both negative expressive intensity (Fig. 4g: $t(226) = -4.2$, $p = 5e\text{-}13$, $r = -0.18$, 95% CI = −0.23 to −0.13, $bf = 9e + 9$) and blast level selections (Fig. 4h: $t(226) = -4.5$, $p = 9e\text{-}6$, $r = -0.29$, 95%

CI = −0.40 to −0.17, $bf = 1802$). These results suggest that a break was effective at reducing negative emotionality and aggression regardless of whether it was forced or voluntary.

**Negative emotion compounds, increasing aggression within couples.** The preceding analysis examined behaviour solely as a function of the intra-personal emotions experienced by the aggressor (the winner of each round). Based on the assumption that inter-personal affective processing also matters, we predicted that negative affective displays would compound, causing aggression to be greater when both partners displayed high levels of negative affect. To test our prediction, the distribution of blast levels in the "immediate response" condition was examined as a function of whether neither, one, or both members of each couple exhibited a high level of negative emotion (see Fig. 5). When neither partner displayed high levels of negative emotion (white) blast levels were generally low. By contrast, there was a significant increase in aggression when either the winner (red: $w = 2735.5$, $p = 5e\text{-}5$, $r = 0.25$, 95% CI = −3.0 to −1.0, $bf = 1e + 4$) or loser (black: $w = 5407.5$, $p = 0.01$,

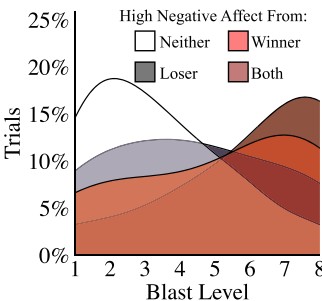

**Fig. 5 | Negative emotion compounds within couples.** To assess the interaction between partners we examined aggressive behaviour (the proportion of responses at each level of blast) as a function of whether neither, one, or both members of a couple expressed *high negative affect*. Density plots from the immediate response condition revealed that when neither player expressed high negative affect (white: *m* = 3.4; *sd* = 2.0) behaviour was dominated by low blast level responses. A significant increase in higher blast level responses occurred when either the loser (grey: *m* = 4.3; *sd* = 2.5) or winner (red: *m* = 5.1; *sd* = 2.6) displayed high negative affect. Furthermore, blast level selections were significantly higher again when both the winner and loser displayed high negative affect (burgundy: *m* = 6.2; *sd* = 2.3). Aggressive behaviour was significantly higher when both members of a couple expressed high levels of negative emotion.

$r = 0.17$, 95% CI = −1.0 to −3e-5, *bf* = 21.53) displayed a high level of negative emotion, reflecting a clear shift towards more aggressive blasts. Notably, however, aggression levels were similar regardless of which member of the pair expressed negative emotion (*w* = 1181, *p* = 0.11, *r* = 0.15, 95% CI = −2.0 – 7e-5, bf = 0.69; though, as our Bayes Factor indicates, we lack credible evidence for or against the null hypothesis). More importantly, a further increase in blast levels occurred when *both* players showed high negative affect (burgundy) compared to neither player (*w* = 1118.5, *p* = 2e-8, *r* = 0.34, 95% CI = −4.0 to −2.0, bf = 5e + 8), the loser only (*w* = 548.5, *p* = 6e-4, *r* = 0.35, 95% CI = −3.0 to −1.0, bf = 47.13), and the winner only (*w* = 455, *p* = 0.047, *r* = 0.24, 95% CI = -2.0 – 2.6, bf = 1.01), demonstrating that negative emotion compounds within couples.

**The dynamics of aggression reveal retaliation, but not escalation, over time.** Based on previous research[28] and theory[10,28], we predicted that blast levels in later rounds would be higher than blast levels in earlier rounds (indicating escalation over time). Counter to our expectations, analysis of blast levels over time revealed no evidence of escalation (see Fig. 6a; *t(1548)* = 1.1, *p* = 0.27, *r* = 0.03, 95% CI = −0.03–0.12, bf = 0.10). Comparison of grand mean blast level data between partners revealed a significant and large correlation however, indicating that partners tended to match each other's overall levels of aggression (see Fig. 6b; *t(56)* = 12.6, *p* < 2e-16, *r* = 0.84, 95% CI = 0.78 – 0.91, bf = 5e + 14). Consequently, we further examined the relationship between each blast level selection (round i) and the following blast level selection (Round i + 1) to assess the dynamics of behaviour. Analysis of single-trial blast level data revealed a significant correlation between each blast and the subsequent blast, providing evidence of immediate retaliation (see Fig. 6c; *t(750)* = 13.3, *p* < 2e-16, *r* = 0.44, 95% CI = 0.38–0.49, bf = 7e + 29). Importantly, the pattern of trial level matching can be seen within individual couples, revealing a striking pattern of matching blast levels between individuals over successive rounds of the game despite clear between couple variability. Data from three separate couples are illustrated in Fig. 6d–f (note that 'win streaks' were collapsed, as outlined in methods), revealing an oscillating pattern of win–loss for each player.

The presence of matching led us to examine patterns of escalation at a trial level, in addition to the game level described above (and shown to be absent in Fig. 6a). For this, we calculated the difference between each player's blast selection (blast on round i ($R_i$)) and their partner's previous blast selection (blast $R_{i-1}$), resulting in escalation/de-escalation coefficients

ranging from a possible −7 to 7. We found that participants chose blast levels higher than their opponent's last selection (i.e., they escalated) on 46% of trials, they de-escalated on 39% of trials, and they matched on 15% (see Fig. 6g). Indeed, within each couple, all three patterns of behaviour were typically visible over time (illustrated in an example couple shown in Fig. 6h). There was, however, notable between-participant variation in escalation tendencies (i.e., many participants never matched their partner exactly, while a small portion matched 75% of the time; as illustrated by the distributions shown in Fig. 6g). Finally, therefore, we also examined the relationship between escalation and de-escalation tendencies within each couple (Fig. 6i), revealing a negative relationship between the escalation tendency of the higher-escalation partner and the lower-escalation partner (*t(50)* = −12.5, *p* < 2e-16, *r* = −0.87, 95% CI = −0.92 to −0.78, bf = 6e + 13). Asymmetry in escalation tendencies within couples suggests that one partner tended to consistently choose blast levels that were higher than their partner's last selection, while the other partner consistently chose lower blast levels, exemplified by the couple shown in Fig. 6h. Nonetheless, as described above (and illustrated in Fig. 6b, c), even with this escalation tendency asymmetry, most couples still matched each other's mean blast levels.

**Putting it all together: modelling a mechanism.** The above analysis has demonstrated that forced breaks reduce negative affect and blast levels (behavioural aggression) under conditions of high negative affect; however, the potential mechanism behind forced breaks can be further illuminated by statistical modelling of the data. Specifically, we employed multi-level, longitudinal Actor-Partner Interdependence Modelling[53,54] (APIM) to examine the interplay of both partner's emotions over time. APIM allows us to assess the impact of the previous winner's and loser's negative affect (at $R_{i-1}$) on the current winner's and loser's negative affect (at $R_i$), remembering that each player oscillates between winning and losing over time (as described above). Specific model parameters and full statistics for all following models are provided in Table 3.

In the Immediate Response condition, the multi-level longitudinal APIM model (Fig. 7a reveals that the current winner's negative affect (at $R_i$) is affected by both their previous negative affect (as a loser, at $R_{i-1}$; an "actor effect"; *t(326)* = 4.6, *p* < 0.0001, $\beta$ = 0.29, 95% CI = 0.16 – 0.29) and their partner's previous negative affect (as the winner/potential aggressor, at $R_{i-1}$; a "partner effect"; *t(326)* = 2.3, *p* = 0.02, 95% CI = 0.02–0.17), suggesting that the winner of each round is influenced by the negative affect of both players (corroborating the compounding effects shown in Fig. 5). Further, because the model is longitudinal, we expect the pattern to oscillate on each iteration (illustrated with a fading continuation of the model with ellipses), potentially explaining the oscillating pattern of matching shown in Fig. 6d–f. Finally, examination of the impact of both player's negative affect on blast selection using multi-level regression further corroborates our findings that both partners' negative affect drives aggression (*t(345)* = 6.57, *p* = 1.8e-10, *r* = 0.33, 95% CI = 0.02–0.42, bf = 4.5e + 7) as shown in Fig. 7b (mean effect shown in red, pair-level illustrated in grey).

We also provide a second model displaying the same relationships for the forced break condition (illustrated in Fig. 7c). As expected, each player's negative affect at $R_i$ influences their negativity at $R_{i-1}$, but there was no significant impact of the $R_{i-1}$ loser's negativity on the $R_i$ winner's negativity in the Forced Break condition. Given that APIM models are non-Bayesian the null finding must be interpreted as a lack of evidence for an effect, rather than as evidence of the absence of an effect. Nonetheless, Bayesian multi-level regression provides extremely strong evidence that the impact of both partner's negative affect does not predict higher blast levels in the forced break condition (*t(1201)* = 0.51, *p* = 0.61, *r* = 0.01, 95% CI = −0.04 –0.07, bf = 0.07), as is illustrated in Fig. 7d.

Finally, to further illustrate the potential mechanism of forced breaks, we conducted a simple mediation analysis[55,56], to examine the direct and indirect impact of the forced break condition on blast level selection (behavioural aggression) through winner negativity. The mediation analysis (Fig. 8; Table 4) shows that forced breaks (in isolation) are predicted to reduce blast level selections by 0.36 on average (*t(1548)* = −2.6, *p* = 0.010,

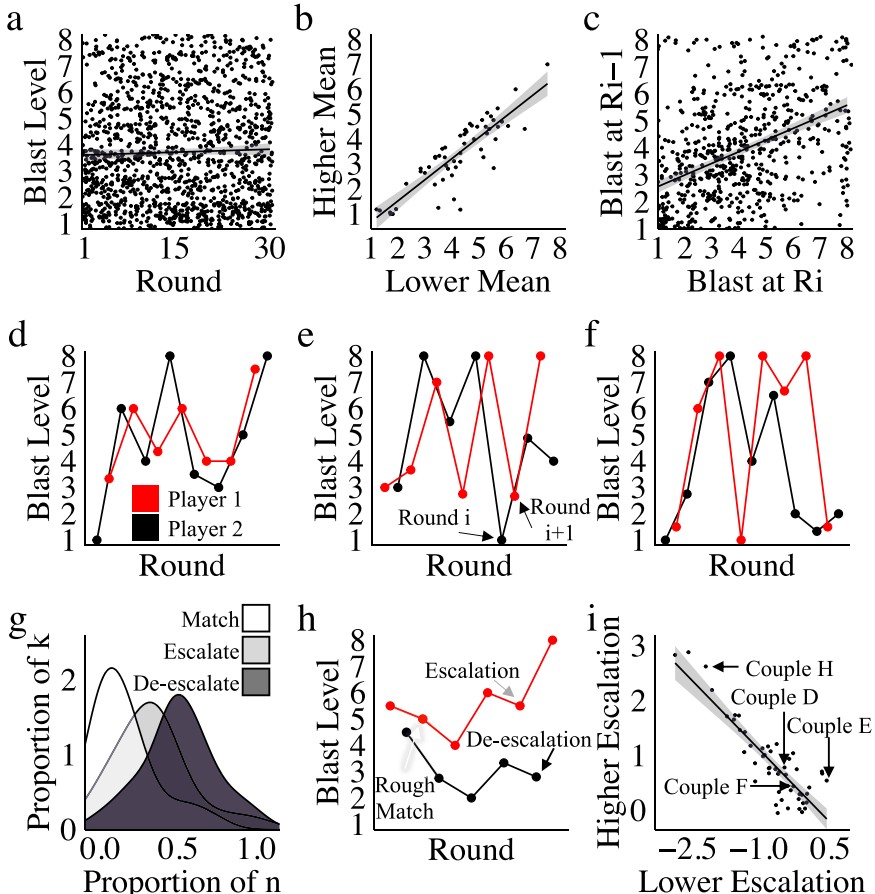

**Fig. 6 | Dyadic conflict behaviour. a** Counter to expectation, results do not reveal a tendency for couples in conflict to escalate over time. **b** When compared directly against each other, however, partners tended to closely match each other's grand mean blast levels. **c** Couples also tended to roughly match each other's blast selections at a trial level, meaning that partners choose blast levels (in R(ound)$_i$) similar to their partner's previous blast level selection (in $R_{i-1}$). **d, e, f** The pattern of blast levels within three separate couples illustrates the matching of blast levels over time. **g** Trial-to-trial changes in aggression were used to calculate the proportion of trials associated with matching, escalation and de-escalation, in each participant, shown as density plots illustrating the prevalence of each behaviour. **h** An example couple's behaviour, showing the dynamics of aggression over time. For each behaviour (matching/escalation/de-escalation) an arrow points to the second of two responses

(one from each player) that exemplify that behaviour. Player 1 (red) consistently makes higher blast level responses than Player 2 (black). Player 1 exhibits both matching and escalation over time, whereas Player 2 only exhibits de-escalation. **i** Comparison of the escalation tendencies within couples reveals a significant asymmetry between escalation and de-escalation when partners are compared against each other. Couples shown in (**d, e, f** and **h**) are identified, illustrating the different patterns of behaviour observed as a function of the escalation/de-escalation asymmetry. *Defining Plot Elements.* Grey shaded regions in regression plots represent standard errors. Boxplot features are as follows: centre line, median; upper and lower box boundaries, Inter Quartile Range (IQR); upper and lower whiskers, smallest or largest value within 1.5× the corresponding IQR; dots, outliers.

$\beta = -0.36$, 95% CI = −0.64 to −0.09) and negativity by 0.11 on average ($t(1548) = -8.73$, $p < 2\text{e-}16$, $\beta = -0.11$, 95% CI = −0.13 to −0.08). When accounting for winner negativity, however, the impact of forced breaks on blast level becomes non-significant ($t(1547) = -1.6$, $p = 0.11$, $\beta = -0.23$, 95% CI = −0.51–0.05) and negativity solely predicts blast ($t(1547) = 4.4$, $p = 1\text{e-}5$, $\beta = 1.26$, 95% CI = 0.70 – 1,82), suggesting a complete mediation effect (i.e., the impact of forced breaks on blast level is entirely dependent on its impact on negativity).

**Replication analysis**

Having established a link between the expression of negative emotion and aggressive behaviour using our face-to-face Competitive Reaction Time Task we carried out a follow-on study (as outlined in the methods) that allowed us to ask whether the effects were replicable. As in the primary study, results of the replication provide strong evidence for the alternative hypothesis[57,58] (Table 5). First, we show again that forced breaks reduce negativity from time one (when the winner is announced) to time two (when the winner picks a blast level; Fig. 9a; $w = 2791$, $p = 2\text{e-}14$, $r = 0.73$, 95% CI = 0.33–0.45, bf = 1.6e + 11). Second, when negativity is high, we see lower blast levels in the forced break condition (Panel b; $w = 34585$, $p = 2.7\text{e-}$

4, $r = 0.12$, 95% CI = 4.5e-5 – 2.0, bf = 52.8). Third, negativity from both players predicts higher blast levels (in a multi-level model, grey regression lines represent the pair-level; Panel c; $t(386) = 3.14$, $p = 0.002$, $r = 0.16$, 95% CI = 0.06–0.25, bf = 12.4). Fourth, players matched each other's blast levels closely (shown on the trial level; Panel d; $t(20) = 9.25$, $p = 1.2\text{e-}8$, $r = 0.90$, 95% CI = 0.77–0.98, $bf = 347396$). In sum, therefore, the follow up study provides a clear replication of our original findings.

Finally, we examined the qualitative responses provided in the follow-up study, as they pertain to our participant's perceptions of the impact of forced breaks on their cognition, emotion, and behaviour. Adopting a simplified thematic analysis approach (cf. Braun and Clarke, 2006[59], 2013[60]) individual participant responses were assessed independently by AGM to identify common and repeated themes within the comments provided, with additional comparison made across individuals to confirm the consistency of the themes. Initial themes were discussed with all authors, leading to a) further refinement, b) the identification of representative quotes to illustrate each theme, and c) a count of the frequency of each theme occurring within the response data. The quotes and frequency counts are presented in Table 6, highlighting the participants keen awareness of responding to their partner's behaviour (matching/retaliation) and feeling uncomfortable about

**Table 3 | Actor-partner interdependency modelling**

| Effect | n | k | Estimate | P value | 95% CI ↓ | 95% CI ↑ | df | Stat |
|---|---|---|---|---|---|---|---|---|
| *Panel a, b* | | | | | | | | |
| P1 Intercept ($R_i$) | 22 | 163 | 0.09 | **0.002** | 0.03 | 0.14 | 326 | 3.1 |
| P1 Actor Effect | 22 | 163 | 0.29 | **<1e-4** | 0.16 | 0.41 | 326 | 4.6 |
| P1 Partner Effect | 22 | 163 | −0.06 | 0.43 | −0.21 | 0.09 | 326 | 2.3 |
| P2 Intercept ($R_i$) | 22 | 163 | 0.23 | **<1e-4** | 0.17 | 0.30 | 326 | 6.9 |
| P2 Actor Effect | 22 | 163 | 0.24 | **0.01** | 0.06 | 0.42 | 326 | 2.6 |
| P2 Partner Effect | 22 | 163 | 0.17 | **0.02** | 0.02 | 0.32 | 326 | −0.8 |
| P1 ($R_i$) ~ P2 ($R_i$) | 22 | 163 | 0.13 | 0.10 | −0.02 | 0.28 | 161 | 1.7 |
| P1 ($R_{i-1}$) ~ P2 ($R_{i-1}$) | 22 | 163 | 0.11 | 0.15 | −0.04 | 0.26 | 161 | 1.4 |
| Blast ($R_i$) ~ Both Negativity ($R_i$) | 23 | 347 | 0.33 | **2e-10** | 0.24 | 0.42 | 345 | 6.6 |
| *Panel c, d* | | | | | | | | |
| P1 Intercept ($R_i$) | 80 | 588 | 0.08 | **<1e-4** | 0.06 | 0.09 | 1176 | 9.3 |
| P1 Actor Effect | 80 | 588 | 0.18 | **<1e-4** | 0.11 | 0.25 | 1176 | 4.8 |
| P1 Partner Effect | 80 | 588 | 0.003 | 0.95 | −0.08 | 0.08 | 1176 | 0.4 |
| P2 Intercept ($R_i$) | 80 | 588 | 0.10 | **<1e-4** | 0.08 | 0.11 | 1176 | 10.4 |
| P2 Actor Effect | 80 | 588 | 0.17 | **2e-4** | 0.08 | 0.26 | 1176 | 3.7 |
| P2 Partner Effect | 80 | 588 | 0.02 | 0.68 | −0.06 | 0.10 | 1176 | 0.1 |
| P1 ($R_i$) ~ P2 ($R_i$) | 80 | 588 | 0.02 | 0.55 | −0.06 | 0.11 | 586 | 0.6 |
| P1 ($R_{i-1}$) ~ P2 ($R_{i-1}$) | 80 | 588 | 0.03 | 0.51 | −0.05 | 0.11 | 586 | 0.7 |
| Blast ($R_i$) ~ Both Negativity ($R_i$) | 81 | 1203 | 0.01 | 0.61 | −0.04 | 0.07 | 1201 | 0.5 |

Bayes Factors for Blast ($R_i$) ~ Both Negativity ($R_i$) in M1 and M2 are 4.5e + 7 and 0.07, respectively. Both models are multilevel, where the pair-level and trial-level variance are included explicitly as parameters, and the participant-level is included implicitly in APIM model design.

doing so (guilt/discomfort). Despite not being asked explicitly to do so, participants also reported awareness of the influence of both their own and their partner's negativity on their behaviour. Given the qualitative nature of this data we do not present any further statistical analysis or make claims based on it alone, using it solely to inform discussion of our principal findings.

## Discussion

We adapted a widely used aggression paradigm (the Competitive Reaction Time Task; CRTT[61]) for use with couples, allowing aggressive behaviour between partners to be examined experimentally. As predicted by the General Aggression Model[10] and the I³ Model[11], we showed that aggression within couples is driven by both inter- and intra-personal experiences of negative emotional arousal, confirming that emotional experiences of both partners can influence an aggressor's actions. Indeed, we observed an 86% increase in mean aggression when both partners were expressing negative emotion compared to when neither partner was doing so. In addition, by examining the dynamic interaction between partners we were able to reveal evidence that couples match each other's aggression over time. Taken together, therefore, these findings substantiate emotional co-regulation (i.e., partners' emotional impact on each other) as a central feature of intimate partner conflict— something long suspected[23,24,62,63], and partially demonstrated[3,20,64,65], but impossible to show with traditional experimental paradigms (i.e., single player or computer opponent versions of the competitive reaction time task).

Most importantly, perhaps, we also showed that the introduction of a short delay (a forced break) between provocation and the opportunity for aggression led to a decrease in aggression. This finding builds on previous research[66], highlighting the possibility of developing practical interventions aimed at reducing intimate partner aggression. Although our findings reveal that forced breaks decrease negative emotion and aggression, we did not find evidence that the length of forced break impacted results (i.e., our 5, 10, and 15 s conditions resulted in similar

reductions in emotional negativity and aggression). Nonetheless, we were able to show that participants who elected to take extra time before responding benefited from a further decrease in negative emotion and exhibited less aggression. This finding is noteworthy because it shows that breaks do not need to be forced to be effective, suggesting that training individuals to take voluntary breaks may be effective as an aggression reduction strategy.

One obvious difficulty for experimental investigations of conflict between couples is the inherent ethical constraint around causing harm, which has often led to the use of relatively innocuous forms of aggression (such as criticizing another person's writing, giving a fictitious opponent hot sauce to eat, and assigning difficult puzzles to a competitor)[67], many of which have been criticized for poor external validity[50]. In the present study, we employed a noxious auditory sound blast (available here: https://shorturl.at/ejvUZ) that was designed to be unpleasant, but not harmful. Whilst our noxious sound is quite unlike acts of physical aggression seen in real-world settings, participants reported it as aversive during debriefing, even expressing fear of receiving blasts and guilt after sending them (see Table 6), suggests that delivering sound blasts is a reasonable experimental proxy for aggressive behaviour. In addition, (rather than examining the presence or absence of aggression) we used participants' choice of blast level (i.e., auditory volume) to provide a variable measure of behavioural aggression.

Here, because we employed a variable measure of aggression, we were able to show that couples closely matched each other's blast levels in both the grand mean and from trial to trial (i.e., if Player 1 selected a blast level of five, Player 2 was likely to select a blast level near five on their next win). Despite providing evidence of retaliation, one clear feature of our findings is the absence of escalation over time, contrary to claims that a Violence Escalation Cycle occurs when two people are in conflict[28]. We note, however, that the present results come from a convenience sample of couples (with relatively low levels of trait aggression), and it therefore remains possible that escalation does occur in couples with histories of severe aggression or abuse. Equally, future studies are required to determine whether factors such as the

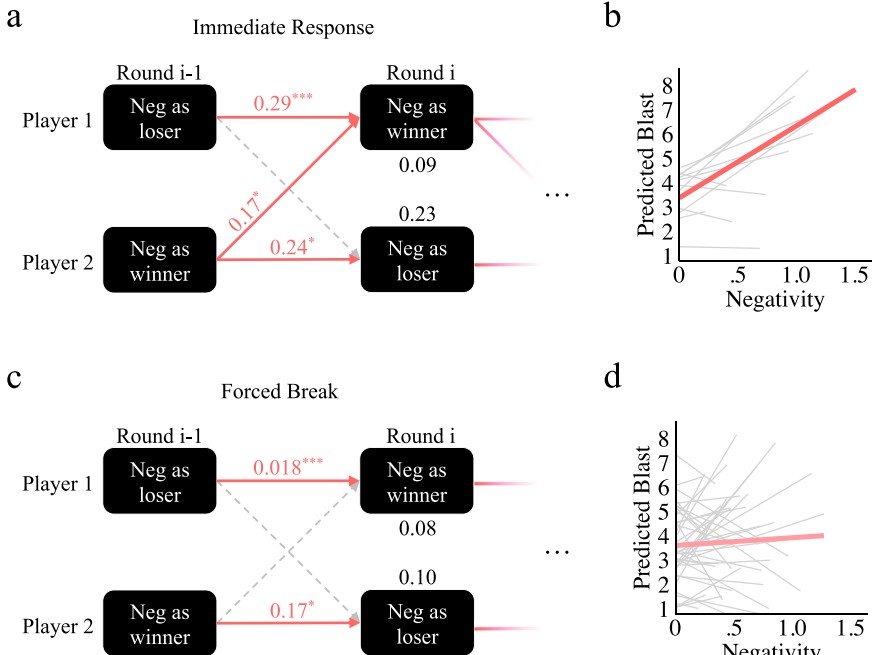

**Fig. 7 | Actor-partner interdependency.** Actor-Partner Interdependence Models allow us to see the impact of each player's emotion on their own, and their partner's, future emotion. These models are longitudinal (illustrated with a fading continuation) and can be read as follows: the black numbers represent the predicted negativity level at time Round i ($R_i$) given that $R_{i-1}$ negativity is zero; each horizontal line represents the "actor effect" (the impact of a given player's negativity on themselves at a later time), and each diagonal line is the "partner effect" (the impact of one's negativity on their partner at a later time); for actor and partner effects, relationships can be read as "for every one unit increase in negativity from [player] at $R_{i-1}$ we expect a unit increase of [number] in [player] negativity at $R_i$," (e.g., in (**a**): for every unit increase in negativity from P1 at $R_{i-1}$, we expect a unit increase of 0.29 in P1

negativity at $R_i$). **a** This model of our immediate response condition shows that the winner of each round (P1 is the winner at $R_i$ in the present model) is influenced by the emotions of both themselves and their partner from the previous round ($R_{i-1}$). **b** Indeed, negativity from both the winner and loser of a given round predict higher blast level selections (grey regression lines show the pair-level of our multi-level analysis). **c** This model of our forced break condition shows that the winner of each round (P1 is the winner at $R_i$ in the present model) is influenced by their emotions from the previous round ($R_{i-1}$). **d** In contrast to the Immediate Response model, the Forced Break model only displays a significant impact of actor effects on future negativity. **d** Along these same lines, regression shows no significant impact of players' emotions on blast level selection (see Table 3 for Bayes factors).

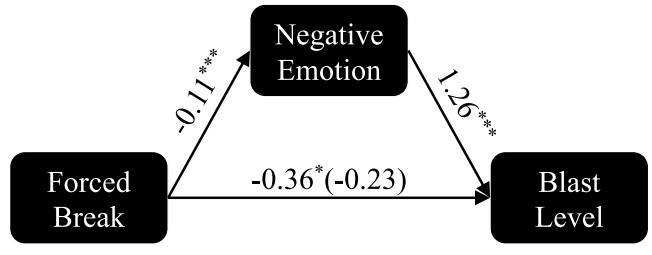

**Fig. 8 | Mediation analysis of the impact of forced breaks on blast selections.** Mediation Analyses allow us to assess the impact of an independent variable on a dependent variable in isolation and when mediated through another independent variable. In the present model, we are testing to see if the forced break condition influences blast level selections directly (bottom arrow) or indirectly (through negativity). Analysis reveals that, in isolation, the forced break condition predicts lower blast levels (by ~0.36 units on average) than the immediate response condition. However, forced breaks also predict lower negativity than the control group (by ~0.11 units on average). When we assess the impact of forced breaks on blast level mediated through negativity, we see that a one unit increase in negativity predicts an increase in blast of about 1.26 units, while the influence of forced breaks directly is not significant (−0.23) in the mediation model. This suggests that the impact of forced breaks on blast is mediated by its impact on negativity (i.e., the mechanism behind the forced break effect likely lies in negativity).

length or seriousness of a relationship, or a previous history of aggressive behaviour, alter the pattern of behaviour reported here. More generally, although the General Aggression Model[68,69] suggest that similar forms of aggressive interactions should occur for all kinds of dyadic relationships, it would be interesting to discover whether the present findings replicate or

differ with other types of dyads (e.g., friends, siblings, teammates, etc.). Regardless, we found that some individuals had a greater tendency to choose blast levels larger than their partner's last blast choice, evidence of an interpersonal asymmetry in escalation tendency at a trial level (see Fig. 9g, h, i), though partners still tended to match each other overall (see Fig. 6b, c). An important avenue for future research will be to investigate how asymmetries in the escalation tendencies within a couple relate to the overall severity of conflict that results.

As well as being limited by ethical constraints around the experimental induction of aggression, previous understandings of couples' conflict has also been limited by the lack of truly dyadic, face-to-face research. Indeed, studies of intimate partner aggression have typically used questionnaire measures administered individually[70], or have employed 'sham' dyadic designs in which participants are falsely told they may select difficult yoga poses[66] for their partner to hold in a separate room, for instance. Previous studies have approached the present design, bringing couples into a lab to play a fake version of the CRTT in separate rooms (usually they are told that their blasts/shock selections will be delivered to their partner, and vice versa, but they are not)[3,65], or bringing strangers into a lab and having them really send each other blasts, but from separate rooms[28]. By examining couples together, face-to-face and in-person, the present study offers a significant methodological improvement to the standard CRTT, increasing its realism while maintaining its experimental control. Although we employed a modest sample size (53 couples in the primary study and 29 in the replication) our approach also allowed us to maximise power by focusing on trial level data (which gave us thousands of observations for most analyses). Critically, we also employed an innovative and underused approach to the analysis of behaviour (discussed below).

## Table 4 | Mediation analysis

| Effect | *n* | *k* | Estimate | *P* value | 95% CI ↓ | 95% CI ↑ | df | Stat |
|---|---|---|---|---|---|---|---|---|
| *Mediation Analysis* | | | | | | | | |
| Forced Break → Blast (Isolated) | 104 | 1550 | −0.36 | **0.01** | −0.64 | −0.09 | 1548 | −2.6 |
| Forced Break → Neg | 104 | 1550 | −0.11 | **<2e-16** | −0.13 | −0.08 | 1548 | −8.7 |
| Neg → Blast | 104 | 1550 | 1.26 | **1e-5** | 0.70 | 1.82 | 1547 | 4.4 |
| Forced Break → Blast (Mediated) | 104 | 1550 | −0.23 | 0.11 | −0.51 | 0.05 | 1547 | −1.6 |

## Table 5 | Full Statistics for Replication Analysis

| Sample | Test | Panel | *n* | *k* | Effect Size | P-value | Bayes Factor | 95% CI ↓ | 95% CI ↑ | df | Stat |
|---|---|---|---|---|---|---|---|---|---|---|---|
| Fig. 9 (Replication) | | | | | | | | | | | |
| Winners with high negativity | Wilcoxon Signed-Rank Test | a | 22 | 55 | 0.73 | **2e-14** | 2e-11 | 0.33 | 0.45 | NA | 2791 |
| | | b | 55 | 874 | 0.12 | **2.7e-4** | 52.8 | 5e-5 | 2.0 | NA | 34585 |
| Full Sample | Least Squares Regression | c | 26 | 778 | 0.16 | **0.002** | 12.4 | 0.06 | 0.25 | 386 | 3.1 |
| | | d | 40 | NA | 0.90 | **1.2e-8** | 3.5e-5 | 0.77 | 0.98 | 20 | 9.2 |

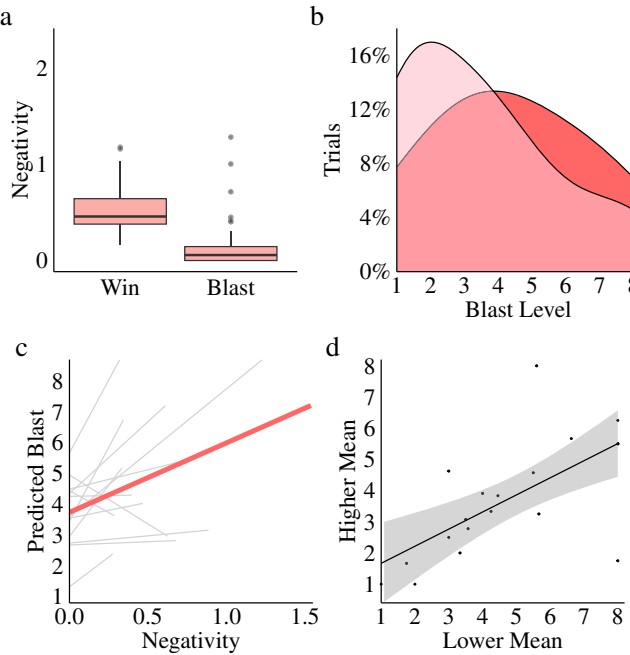

**Fig. 9 | Replication of the primary study.** A follow-up to the original study replicated key findings. Specifically, **a** In the forced break condition, we observe a significant decrease in negative emotion from the time the winner is announced to the time the winner selects a blast level (10 s later). **b** Under conditions of *high negative emotion*, blast selections are lower in the forced break condition than the immediate response condition. **c** Negative expression from both players predicts higher blast level selections from the winner. **d** Players closely matched each other's overall mean blast levels, suggesting retaliation as shown in the primary study. Defining Plot Elements. Grey shaded regions in regression plots represent standard errors. Boxplot features are as follows: centre line, median; upper and lower box boundaries, Inter Quartile Range (IQR); upper and lower whiskers, the smallest or largest value within 1.5x the corresponding IQR; dots, outliers.

Rather than examining aggressive behaviour in isolation, we recorded participants' facial expressions, which allowed behaviour to be examined as a function of trial-to-trial changes in emotional arousal, illuminating the dynamics of dyadic affective and behavioural processes during conflict. This allows us to identify and examine both essential components of reactive aggression, behavioural aggression and negative emotional arousal, rather than assuming reactive aggression from behaviour alone (a method used successfully in previous studies[71]). To our initial surprise, assessment of the emotional expressions elicited during performance of the task highlighted that participants were most often experiencing positive emotion (i.e., happiness) suggesting that they enjoyed their participation. Nonetheless, by combining high time-resolution affect coding and selective averaging procedures, we were able to show that high levels of negative affect (indexed by prototypic displays of anger and disgust[20,21,51,52]) predict increased levels of aggression, regardless of which member of the couple was expressing the negative emotion.

Given the dyadic nature of conflict, it has long been theorized that an aggressor's behaviour is influenced both by negative emotions in the aggressor[62,63], and by the aggressor's perception of their target's emotion[10,24,28,62,64,72]. Our finding that the emotions of both partners impacts the behaviour of an aggressor is notable because it provides experimental support for this claim (rather than relying on retrospective self-reports or observation), adding to a growing body of evidence highlighting the importance of emotional co-regulation as a driver of aggression[3,16,20,23,24,64,72]. For example, two recent studies using face-to-face dyadic designs found that partners' negative emotional expressions correlated with each other[23] and with aggressive behaviour during conflict[20]. The present findings are distinctive, however, in showing that, as well as compounding between individuals, aggression is also more likely when only the non-aggressing member of a couple expresses negative emotion. One implication of this finding is that an individuals' aggressive behaviour can be influenced by the emotional expression of those they interact with. For example, if the compounding of emotion seen here extends beyond dyadic interactions to larger social groupings, the present results may help explain why violence can erupt so readily within crowds[73–75]—where an individual may be simultaneously exposed to multiple expressions of negative emotion. We stress, however, that within the context of intimate partner aggression, it is important not to misinterpret our findings as suggesting that a victim/target of aggression is responsible for the behaviour of the aggressor (cf. victim blaming[76]). Nonetheless, a co-regulatory account of intimate partner aggression does suggest that each partner within a couple will influence the other. In particular, the current findings suggest that effectively regulating one's own negative emotion should be an effective strategy for managing conflict in both one-self and one's partner.

More broadly, the present findings confirm that it is possible to reduce aggression within intimate dyadic interactions. Here, we used short involuntary time delays (of 5, 10, or 15 s) to influence behaviour, revealing a marked reduction in negative expression and aggressive behaviour following forced breaks (of all lengths), and we were able to

**Table 6 | Themes and quotes from post-experimental survey**

| Theme | n | Representative Quote(s) |
|---|---|---|
| *Main Themes* | | |
| Matching/Retaliation | 43 | "I felt annoyed when I saw [my partner] had chosen a high number for no apparent reason. My [blast] level was definitely determined by [my partner's] previous one if I felt it was too high." |
| Guilt/Discomfort | 29 | "[I] felt bad about choosing a higher level after thinking about it for some time [during the break]."; "[…We] saw in the first round how uncomfortable the noise was so I wouldn't [want to] subject [them] to that." |
| Influence of own negativity | 17 | "I [got] a little bit angry when my partner [sent me a] high blast level after I [kept] sending low levels, so I [would] immediately send [a] high [blast] level [in] response." |
| Mention of Provocation | 14 | "[I was] totally [influenced by their last blast]. [I] felt great after winning and when I lost or if [my partner] blasted me above 3, I wanted to institute a sex ban." |
| Impact of Forced Breaks | 9 | "[With no breaks], the more I lost and the more I got blasted, the higher the number I would give. [With the breaks], I remembered that I love this person and I don't need to prove anything."; "The break tended to reduce any heightened feelings." |
| Influence of partner negativity/ emotion | 7 | "[My] blast level was determined by [my partner's] previous [blast selection] and by their most recent reaction to winning." |
| *Extended/Additional Quotes* | | |
| "I would usually decide which [blast level] I would do based on instinct but then second guess myself during the break" | | |
| "Initially, after losing many in a row, I was going to press a higher number [but] then I realised that my partners response will most likely be the same and so to protect myself I'd lower the level. I also felt bad about choosing a higher level after thinking about it for some time [during the break]." | | |
| "[My blast] was definitely determined by my partner's previous response; if [they] had been a bit cocky or had given me a loud blast, I felt it was more justified for me to give [them] a loud blast as well. The forced break made me look into [their] eyes more and be reminded of the love between us, so I often felt like I just wanted to give [them] a [blast level] 1." | | |
| "Without the breaks it felt very time dependent, rushed, and very competitive so I mostly sent [5-8 s]. In comparison to the breaks which allowed me to realise that [my opponent] is actually my partner and I shouldn't be destroying [their] eardrums." | | |

Quotes have been gender neutralized.

replicate these findings in a follow-on study. Within the theoretical aggression literature[10,11], it has been posited that both inter- and intra-personal affective processes, as well as impulsivity[3,8], impact aggression. Based on the current findings we propose that forced breaks reduce aggression by first blocking the expression of impulsive behavioural aggression while participants are in a provoked state. Then, because negative affect reduces over time, when a response is expressed, it will tend to be less aggressive (assuming no further provocation has occurred). We expect, of course, that a range of other factors, including individual traits and situational variables (such as the couple's escalation/ de-escalation asymmetry), will also shape the outcomes of couples' conflict. Nonetheless, in practice, our results provide strong evidence for the role of impulsivity and negative affect in intimate partner aggression, at least during the kind of short-term, acute conflict present in our experiment.

## Limitations

Although we believe that the findings of the present studies are exciting, it is important to recognise limitations. As mentioned, because we only studied couples in the experiments reported here, we don't know how these results would generalise to other forms of dyad, such as friends or team-mates. Fortunately, future studies using our Face-To-Face version of the CRTT can examine how factors such as the type of relationship or the gender of each partner influences aggression within couples. We also used a convenience sample of mostly undergraduates, who, as expected, scored relatively low on our measure of trait aggression (and although our follow-up study contained a larger age range it was still limited in its diversity). Relatedly, use of undergraduates also defined the nature and length of the relationships between our couples. Future work could usefully examine couples more broadly (e.g., longer-term couples, couples who are married or co-habiting, etc.).

A common critique of the CRTT is that there is no non-aggressive response option. Participants must give a sound blast, which makes it difficult to infer aggressive intent from the use of a sound blast. We chose our response format based on recommendations from previous work[35], and while participants could not choose "no response", in practice they did use the full range of blasts available to choose how much discomfort to inflict. Nonetheless it is important to consider the fact that when inflicting discomfort is facilitated (by design) by an experimental setup, it limits the extent to which we can make claims about "real-world" aggression. In addition, it could also be valuable to include a trait measure of negative urgency in future studies on forced breaks, as variability in person level variables between participants might moderate the effect of forced breaks on negative affect and aggression. Equally, it is important to recognise that intimate partner aggression is not a unitary phenomenon and our focus on the role of negative emotion and impulsivity means that our results can only speak to some forms of aggression between partners. Specifically, forced breaks can only be effective in reducing reactive aggression. Forced breaks should not be expected to have any influence on the systematic, premeditated forms of abuse used to exert long-term control over a partner.

## Conclusion

The present study involved an innovative task design, alongside physiological and behavioural measures, and focused trial-level analyses, which together allowed us to demonstrate that forced breaks can successfully reduce aggressive behaviour between couples. This finding is noteworthy for highlighting a potential route to intervention, even for couples where both partners express high levels of negative affect. For example, practitioners providing support for couples may be able to train couples in the use of short breaks as a technique for managing conflict. As well as investigating the possible application of forced breaks in applied contexts, one important aim for future research will be to assess whether fully voluntary breaks (i.e., internally counting to ten) in isolation are as effective as forced breaks (i.e., an externally imposed delay) at reducing negative urgency and aggression. Fundamentally, therefore, we are optimistic about the implications of the current findings. Experimental research involving couples can reveal the complex dyadic processes influencing intimate partner aggression[20,65,71,77], adding to a growing awareness of the importance of negative emotional arousal and emotional co-regulation as key drivers of aggression, knowledge that should ultimately contribute to the development of effective interventions.

## Data availability

De-identified data available at https://doi.org/10.5281/zenodo.12936112. Raw video/audio data is not available to protect participant anonymity.

## Code availability

Data were analyzed in R and the analysis code is available along with the data at the DOI provided above. Custom code to run the experiment was written in Python and is available on GitHub (https://github.com/AnnahGrace01/FTF-CRTT), and archived at https://doi.org/10.5281/zenodo.12797202. FACS coding was run through the terminal using OpenFace; detailed information and instructions for running OpenFace can be found on the OpenFace GitHub page.

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

## Acknowledgements

The project leading to these results has received doctoral student funding from the University of St Andrews. The funders had no role in study design, data collection and analysis, decision to publish or preparation of the manuscript. This project was made possible through help from staff in the IT workshop in the School of Psychology and Neuroscience at the University of St Andrews. Further, data was collected with help from two senior honours undergraduate students: Becky Osmond and Maria Bitar.

## Author contributions

Annah G. McCurry: Conceptualization, Methodology, Data collection, Formal analysis, Writing—original draft, Writing—review & editing, Visualization.

Robert C. May: Conceptualization, Methodology, Writing—review & editing, Supervision, Project administration, Funding acquisition. David I. Donaldson: Conceptualization, Methodology, Writing—review & editing, Visualisation, Supervision, Project administration, Funding acquisition.

## Competing interests

The authors declare no competing interests.
