## [Peer Review File · Communications Psychology]

25th Oct 23

Dear Mx McCurry,

Thank you for submitting your manuscript titled "Both Partners' Negative Emotion Drives Aggression During Couples' Conflict" and thank you for your patience in awaiting an initial decision. I have now had the opportunity to read your work and discuss it with the other editors in the Communications Psychology team. Regretfully, we cannot publish this submission in Communications Psychology.

It is our policy to decline a substantial proportion of manuscripts without sending them to referees so that they may be sent elsewhere without further delay. Such decisions are made by the editorial staff when it appears that papers do not meet the criteria for publication in Communications Psychology.

We appreciate your study, which studies aggression in romantic couples.

At this stage, we consider whether the work represents an advance of significant relevance to scientists working in the same area of psychology; whether the data appear appropriately substantive and the analysis suitable to address the research question; and whether there is sufficient evidence in support of the conclusions.

In this case, I am sorry to say that we are not persuaded that the study has sufficient methodological strength given the lack of a control group (e.g., friends, strangers) judged against the journal's criteria to qualify for publication in Communications Psychology. Please note that we do not question the validity of your work.

Although we cannot offer to publish your manuscript, we believe the Editorial Board at Scientific Reports will find it interesting and recommend you transfer there. To transfer your manuscript there, please use our manuscript transfer portal. You will not have to re-supply manuscript metadata and files, unless you wish to make modifications. For more information, please see our manuscript transfer FAQ page.

I am sorry that we cannot offer further consideration for publication in Communications Psychology and thank you for your interest in publishing with us.

Best regards,

Antonia Eisenkoeck

Antonia Eisenkoeck

Senior Editor

Communications Psychology

7th Dec 23

Dear Mx McCurry,

Thank you for your correspondence asking us to reconsider our decision on your Article, "Both Partners' Negative Emotion Drives Aggression During Couples' Conflict" and apologies for the delay in getting back to you. After careful consideration we have decided that we would be willing to consider a revised version of your manuscript and send this out to review.

When revising your paper please:

- ensure that it complies with the editorial policies outlined below

- ensure that the statistics reporting and interpretation are in line with journal guidelines <https://www.nature.com/commpsychol/submit/submission-guidelines#statistical-guidelines> there are multiple instances where a non-significant finding derived from null hypothesis significance testing, i.e. absence of the evidence for an effect, is used as evidence of the absence of an effect. Instead, please provide Bayesian statistics that offer actual evidence for the absence of an effect.

Moreover, if you wish to have your manuscript peer-reviewed anonymously, please ensure that the data deposition does not allow the tracing of your identity. The data deposition leads to your university and can be traced to <https://research-portal.st-andrews.ac.uk/en/datasets/ftf-crtt-dataset> which lists your names. We commend your data sharing intentions, however, unless the reviewers ask for the data, it is permissible to not include a link to the deposition until potential acceptance of the paper. If the reviewers ask for the data and code, and you want to maintain anonymity, we can advise you on platforms that will allow anonymous data sharing.

Thank you for raising the requirement to deposit the author version of the accepted paper. Please be aware of the following stipulations that pertain to manuscripts once they are accepted for publication:

Acceptance of the manuscript is conditional on all authors' agreement with our publication policies (<https://www.nature.com/commpsychol/editorial-policies>). In particular, your manuscript must not be published elsewhere and there must be no announcement of the work in the media until the publication date.

Publication is typically within two to three weeks of acceptance. Between acceptance and publication, you may wish to make your institution's press office aware of the forthcoming publication, if you wish to bring your work to the media's attention, so that they can start preparing any publicity. Please note that the paper is still under embargo until it is published in the journal. Further details of our embargo policy can be found here <http://www.nature.com/authors/policies/embargo.html>.

To resubmit your revised manuscript, please use the following link:

[link redacted]

Best regards,

Antonia Eisenkoeck

Antonia Eisenkoeck

Senior Editor

Communications Psychology

EDITORIAL POLICIES AND FORMATTING

Editorial Policy: Policy requirements (Download the link to your computer as a PDF.)

Furthermore, please align your manuscript with our format requirements, which are summarized on the following checklist:

Communications Psychology formatting checklist

and also in our style and formatting guide Communications Psychology formatting guide .

* **CODE AVAILABILITY:** All Communications Psychology manuscripts must include a section titled "Code Availability" at the end of the methods section. In the event of publication, we require that the custom analysis code supporting your conclusions is made available in a publicly accessible repository; please choose a repository that provides a DOI for the code; the link to the repository and the DOI must be included in the Code Availability statement. Publication as Supplementary Information will not suffice. We ask you to prepare and upload code at this stage, to avoid delays later on in the process.

24th Apr 24

Dear Mx McCurry,

Thank you for your patience during the peer-review process. Your manuscript titled "Both Partners' Negative Emotion Drives Aggression During Couples' Conflict" has now been seen by 2 reviewers, whose comments are appended below. You will see that they find your work of some potential interest. However, they have raised quite substantial concerns that must be addressed. In light of these comments, we cannot accept the manuscript for publication, but would be interested in considering a revised version that fully addresses these serious concerns.

We hope you will find the Reviewers' comments useful as you decide how to proceed. Should additional work allow you to address these criticisms, we would be happy to look at a substantially revised manuscript. If you choose to take up this option, please highlight all changes in the manuscript text file, and provide a detailed point-by-point reply to the reviewers.

Editorially, we ask that you remove any inferences on intimate partner violence as these go beyond what the present study tested. As you can see from the reports, the reviewers see merit in testing for the mechanism through which a delay reduces aggressive action in the task, but both experts find the mechanistic insights too preliminary. In revision, we ask you to address this shortcoming; as Reviewer #1 highlights, this will likely require new empirical data.

I am also attaching a checklist that details critical reporting requirements for the revised manuscript. Please attend to each item and ensure your manuscript is fully compliant. We are requesting that your manuscript aligns with these requirements as this facilitates the evaluation of your manuscript, reducing delays in re-review and potential future acceptance. If your revised manuscript is not aligned with these requests on major issues, such as those concerning statistics, it may be returned to you for further revisions without re-review. Additional information can be found in our style and formatting guide Communications Psychology formatting guide.

If the revision process takes significantly longer than five months, we will be happy to reconsider your paper at a later date, provided it still presents a significant contribution to the literature at that stage.

Please use the following link to submit your

- revised manuscript,
- point-by-point response to the referees' comments,
- cover letter (as a separate document),
- the Editorial Policy Checklist (see below),
- the Reporting Summary (see below), and
- the completed Editorial Request Table (attached):

[link redacted]

Thank you for the opportunity to review your work.

Best regards,

Marike

Marike Schiffer, PhD

Chief Editor

Communications Psychology

REVIEWER EXPERTISE:

The reviewers share expertise in research on intimate partner violence and lab-based studies / dyadic games eliciting aggression

REVIEWER REPORTS:

Reviewer #1 (Remarks to the Author):

I have now had an opportunity to review the manuscript “Both Partners’ Negative Emotion Drives Aggression During Couples’ Conflict” for possible publication at Communications Psychology. This study looked at unprovoked aggression within couples. The authors found that displays of negative emotion (inferred from facial expressions) was associated with greater aggressive responding on subsequent trials and that delayed responding was associated with less aggressive responding.

There are many positive aspects of the current study. The use of existing dyads is under-used in aggression research, they measured actual behaviors, the data analyses are complete, the figures are top-notch, and the addition of looking at facial expressions is clever. Nevertheless, I am less enthusiastic about this study as a single study publication.

A “big picture” concern I have is whether the methods created a condition where you would actually expect aggression to occur. It might be helpful for the authors to look at the I-cubed literature. This literature might help them explain some of their findings (e.g., why there was no association between trait aggression and CRTT performance, why there were generally low levels of overall aggression, etc.) and determine whether they created conditions where they would actually expect aggression. Given that there was no instigation or provocation before completing the CRTT, it doesn’t seem too surprising that aggression was generally low in the current study. It is commonly found that there is not much of an effect of factors such as cognitive resource depletion on aggression without an instigating event (e.g., Finkel et al., 2009: 40_FinkelDewallSlotterOatenFoshee2009_JPSP.pdf (squarespace.com)). Likewise, although prior research has shown that the Buss-Perry Aggression Questionnaire is not strongly associated with the CRTT (e.g., Chester et al., 2019; Sci-Hub | Validating a Standardized Approach to the Taylor Aggression Paradigm | 10.1177/1948550618775408 (gupiaoq.com)), trait aggression predicts intimate partner violence when there is an instigation (e.g., Finkel et al., 2012; Using_I-3_Theory_to_Clarify_When_Disposi20160520-14948-1hug37o-libre.pdf

(d1wqtxts1xzle7.cloudfront.net)). Simply put, if you just put (presumably happy or at least not actively fighting) couples in a laboratory together, they probably won't be too aggressive if there hasn't been any instigation or provocation. Why would they be aggressive without any reason?

Even though there was no instigation in the current study, there were still interesting findings. One of the clearest findings from the current study is that delaying responding was associated with less aggressive behaviors. This finding replicates prior research (e.g., Finkel et al., 2009). When discussing their delayed responding analyses, the authors imply they are testing competing hypotheses about delayed responding and rumination. For example, in their Discussion section, the authors say that "rumination should increase during a break period, leading to an increase in negative emotional arousal and aggression, whereas negative urgency is blocked by a forced break, theoretically leading to reduced aggression." They say something to this effect in their Introduction section too. I am not sold on the idea that the current study says much about the relationship between rumination and aggression. With rumination, it is not the mere passage of time, but it is the active processing and maintenance of provocation-related thoughts (e.g., Bushman et al., 2005; document (psu.edu)). For the results to be relevant to rumination, you'd actually need to know what is happening when people are delaying their responses. Are they simply stepping away and cooling down (not rumination)? Are they distracting themselves (also not rumination)? Or are they stewing about an instigation (rumination)? To test hypotheses about rumination, therefore, the authors would actually need to measure whether people are ruminating or instruct some people to ruminate (and instruct others not to ruminate) or something. You cannot simply equate delayed responding with rumination because those are not the same thing.

In all, I think this study is well done. I am not sure if there is a clear enough take-home message for this to be a stand-alone study though. To make stronger claims about intimate partner violence, I feel like there needs to be better conditions for aggression to occur (e.g., an instigation). And for there to be stronger inferences about why delayed responding works, I feel like there would need to be methods that are designed to assess why delayed responding leads to less aggression.

Reviewer #2 (Remarks to the Author):

This paper reports the results of a study examining the influence of displayed affect and forced breaks during a competitive reaction time task in couples. The results showed that participants were more aggressive in terms of selecting louder noise blasts when they or their partner were expressing negative affect. The result also showed that creating a pause between announcing the trial winner and selecting a noise blast decreased aggression. This study uses novel methods (face-to-face competitive reaction time task, multi-modal assessment of aggression and affect) and has

the potential to add to the field. I do have a few suggestions for the manuscript to have its maximum impact.

I had some questions about the theoretical justification for the study. The authors contrast negative urgency and rumination as factors involved in couples' conflict/aggression. Specifically, they say that a break would reduce the effect of negative urgency but accentuate the effect of rumination. I know that previous research has clearly distinguished trait and state rumination (e.g., Nolen-Hoeksema et al., 2008), so I think rumination could make sense here (although the idea of ruminating for 5-15 seconds may not make sense), but I think of negative urgency as a trait. That is people high in negative urgency behave impulsively when they are experiencing a state of negative emotion. Along these lines, a recent study found that negative urgency was positively related to aggression on the Taylor Aggression Task when participants were displaying greater negative affect (as assessed with FACS) but not when they were displaying less negative affect. Based on this, I'm not sure that I understand the authors' argument that negative urgency would take charge with no break. I would see it as affect would be driving that state affect but perhaps for people high in negative urgency. I also found it slightly confusing that you talk about negative urgency and rumination, but then they aren't measured or manipulated in the study. I guess these suppositions would better fit in the discussion given the study design.

I had a few comments on the methods. I would like to see the internal consistency of the measures. It's important to show that the measures are reliable.

From what I can tell, the authors did not take into account that the data came from couples who were face-to-face. I would expect some kind of multi-level modeling to account for the covariation in the data. I would also think that actor-partner modeling would be of interest because this would allow you to see how one partner's affect display is correlated with the other partner. Y.

It seems that the authors did not include a zero or no noise blast. This makes it more difficult to interpret noise blast selection as aggression since there is no non-aggressive response. This limitation should be noted.

It may be because the method section is later in the document, but I felt like I needed a data analysis section to link the analyses to the research questions. The results started with several analyses that I was not expecting based on the introduction and it decreased the readability.

I think the authors did not include/review some relevant studies. The authors are correct that this is the first face-to-face couples aggression study, but there are several studies that have used aggression paradigms to look at couples' aggression. For example, Eckhardt et al. (2021) found that

alcohol-induced aggression ostensibly against a partner was mediated by cognition, not affective expression. As noted above, Bresin et al., (2022) examined negative urgency and negative affect facial expression in laboratory aggression ostensibly against a partner. Watkins et al (2015) manipulated rumination versus reappraisal in a couples version of the Taylor aggression paradigm. The current study adds to this work, but this work seems relevant in interpreting the current results.

EDITORIAL POLICIES

We ask that you ensure your manuscript complies with our editorial policies and reporting requirements.

To that end, we require revised manuscripts to be accompanied by two completed items: a reporting summary that collects information on study design and procedure, and an editorial policy checklist that verifies compliance with all required editorial policies

Nature Research Reporting Summary

Editorial Policy Checklist

All points on the policy checklist must be addressed. Your revised manuscript can only be sent back to the referees if these checklists are completed and uploaded with the revision.

Notes: If you have submitted a Stage 1 Registered Report, Review, Primer, Comment, or Perspective you do not need to submit these forms. If you have already submitted these forms, you may disregard this request.

* **TRANSPARENT PEER REVIEW:** Communications Psychology uses a transparent peer review system. This means that we publish the editorial decision letters including Reviewers' comments to

the authors and the author rebuttal letters online as a supplementary peer review file. However, on author request, confidential information and data can be removed from the published reviewer reports and rebuttal letters prior to publication. If your manuscript has been previously reviewed at another journal, those Reviewers' comments would not form part of the published peer review file.

Comments from Reviewer 1:

1. I am less enthusiastic about this study as a single study publication.

We appreciate this caution and agree that multi-study publications have greater rigour than single-study publications (all else being equal). To improve our manuscript, we have included data from a follow up study that replicates key findings. Specifically, the replication results state that:

“First, we show again that forced breaks reduce negativity from time one (when the winner is announced) to time two (when the winner picks a blast level; Figure 9a; $w=2791$, $p=2e-14$, $r=0.73$, $95\% CI=0.33-0.45$, $bf=1.6e+11$). Second, when negativity is high, we see lower blast levels in the forced break condition (Panel b; $w=34585$, $p=2.7e-4$, $r=0.12$, $95\% CI=4.5e-5-2.0$, $bf=52.8$). Third, negativity from both players predicts higher blast levels (in a multi-level model, grey regression lines represent the pair-level; Panel c; $t(386)=3.14$, $p=0.002$, $r=0.16$, $95\% CI=0.06-0.25$, $bf=12.4$). Fourth, players matched each other’s blast levels closely (shown on the trial level; Panel d; $t(19)=3.14$, $p=6.7e-4$, $r=0.68$, $95\% CI=0.35-0.86$, $bf=41.5$).” (Page 22)

To illustrate the replication findings, we have introduced a new figure (9) that allows us to show the critical main findings from our original study (limiting reporting to key findings for the purpose of brevity).

Further, adding the replication study also allowed us to gather additional qualitative data that speaks to the mechanism of forced breaks (i.e., thematic analysis of participant free-text responses about their thoughts/feelings/actions during the forced breaks). These data are reported in the results section, with reference to a new table (6), revealing:

“participants keen awareness of responding to their partner’s behaviour (matching/retaliation) and feeling uncomfortable about doing so (guilt/discomfort). Despite not being asked explicitly to do so, participants also reported awareness of the influence of both their own and their partner’s negativity on their behaviour. Given the qualitative nature of this data we do not present any further statistical analysis or make claims based on it alone, using it solely to inform discussion of our principal findings.” (Page 23)

Although we have been careful not to over-interpret these qualitative data we hope that the reviewers will agree with us that they provide rich insight into our couples’ experience that help inform our understanding of the quantitative data patterns.

2. A “big picture” concern I have is whether the methods created a condition where you would actually expect aggression to occur. It might be helpful for the authors to look at the I-cubed literature. This literature might help them explain some of their findings (e.g., why there was no association between trait aggression and CRTT performance, why there were generally low levels of overall aggression, etc.) and determine whether they created conditions where they would actually expect aggression.

We appreciate this concern and agree with the reviewer that instigating aggression inside the lab is always a difficult and delicate task. However, we argue that our methods do have instigation build into the task, by design, according to the I³ model, and as supported by our empirical findings. In response to the reviewers concern we have now made this argument explicit in the introduction, including a new figure (2) explaining our model according to the I³ literature (expanded on further below). The new text states:

“Intimate Partner Aggression is widespread, with around a third of dating couples at universities engaging in some form of physical aggression during conflict⁹. As a result, researchers and practitioners are increasingly focused on understanding the social and emotional interactions that occur between partners during everyday conflict, and how these predict the escalation towards aggression¹⁰. Aggression between partners is frequently reactive in nature – that is, it is impulsive^{e.g., 11,12}, and is characterised by strong negative emotional arousal^{4,5,13}. Critically, emotional arousal and impulsivity are linked, such that acute negative affect (e.g., an angry state) is associated with failures of self-control. As a personality trait, the tendency to act rashly when upset is called ‘negative urgency’^{e.g., 4,5,11,13,14}, and it has been implicated as a central contributor to intimate partner aggression. By contrast, the systematic pattern of abuse sometimes known as ‘intimate terrorism’ or ‘coercive controlling violence’ is less strongly associated with impulsivity and we therefore use the term ‘intimate partner aggression’, rather than ‘intimate partner violence’, to describe our research focus¹⁵.

Within dyads, theoretical accounts such as the General Aggression Model^{e.g., 6} and the I³ model^{e.g., 7} further argue that emotions from both one’s self (*intra*-personal) and one’s partner (*inter*-personal) influences one’s aggressive behaviour. Consequently, if reactive aggression between couples requires negative arousal (influenced by both partners) and impulsivity, we reasoned that it should be possible to reduce behavioural aggression by experimentally blocking impulsive action while participants are in a provoked state. In principle, impulsive action can be prevented by introducing a ‘forced break’ period (e.g., a brief experimentally imposed delay) between provocation and the opportunity for aggression. If impulsivity during a provoked state is critical, we would expect a forced break to limit aggression by blocking impulsive action and reducing negative emotion. In essence, assessing the consequences of an experimentally imposed forced break on the dynamics of dyadic conflict provides a strong test of the influence of both negative affect and impulsivity during provocation, allowing us to examine current accounts of Intimate Partner Aggression.”

(Page 3-4)

In sum, we are very happy to include the I-cubed literature as suggested. Note that we respond to the reviewers comments about the lack of association between trait aggression and CRTT performance, and the overall levels of aggression within our paradigm in more detail below in response to Reviewer 1, point 3.

3. **Given that there was no instigation or provocation before completing the CRTT, it doesn’t seem too surprising that aggression was generally low in the current study. It is commonly found that there is not much of an effect of factors such as cognitive resource depletion on aggression without an instigating event (e.g., Finkel et al., 2009: 40_FinkelDewallSlotterOatenFoshee2009_JPSP.pdf (squarespace.com)). Likewise, although prior research has shown that the Buss-Perry Aggression Questionnaire is not strongly associated with the CRTT (e.g., Chester et al., 2019; Sci-Hub | Validating a Standardized Approach to the Taylor Aggression Paradigm | 10.1177/1948550618775408 (gupiaoq.com)), trait aggression predicts intimate partner violence when there is an instigation (e.g., Finkel et al., 2012; Using_I-3_Theory_to_Clarify_When_Disposi20160520-14948-1hug37o-libre.pdf (d11wqtxts1xzle7.cloudfront.net)). Simply put, if you just put (presumably happy or at least not actively fighting) couples in a laboratory together, they probably won’t**

be too aggressive if there hasn't been any instigation or provocation. Why would they be aggressive without any reason?

We agree that a lack of provocation would be a serious limitation of any study focused on aggression. Fortunately, as noted above, we believe it is clear that we *do* have instigation in the current study; indeed our face-to-face CRTT task was specifically designed to produce conditions that generated provocation and instigated aggressive behaviour. We believe that the behavioural patterns of data in our study show this; in addition, the new qualitative reports we have introduced also strongly suggest that participants felt that this was the case (see table 6). We have now clarified this in the introduction, where the text states:

“In experimental terms, our paradigm includes the features that are expected to produce behavioural aggression according to the I³ model and Perfect Storm Theory¹⁵ (instigation/provocation, an impellent, and dis-inhibition, see Figure 2). Together with affect coding, therefore, our paradigm presents a relatively ecologically valid lab-based dyadic aggression task involving real couples competing and aggressing face-to-face. In particular, both partners can be influenced by each other's negative emotions and each individual can see the direct impact of their (real) behavioural aggression on their partner (as in a naturalistic conflict). Finally, in addition to our primary study, we also present a replication that provides additional support for the initial findings and includes qualitative reports of the impact of forced breaks on our participants, illuminating the mechanism behind forced breaks.” (Page 7)

Given the importance of this issue we have also highlighted it in a new figure (2; as noted above in response to Reviewer 1, point 2).

Further, we think it is important to highlight that we did not see any less behavioural aggression in our experiment than has been reported in previous CRTT studies. In fact, our mean CRTT blast selections are comparable to those reported in the Chester et. al., (2019) study that Reviewer 2 mentions, as well as other validation studies (e.g., Elson et al., 2014). Our argument regarding aggression is that not all trials can reasonably be regarded as aggressive due to a) low blast levels, b) low negative affect, or c) a combination thereof. Thus, the key innovation offered in our analysis (which allows us to focus selectively on aggression when it is present) is the integration of the CRTT with high time-resolution affect coding, which allows us to distinguish aggressive from non-aggressive trials. That is, our analytic approach (using selective averaging techniques borrowed from neuroimaging methods) allows analysis to focus only on the meaningful response trials within the experiment.

On this basis we argue that the CRTT as typically used is a low-yield method (which is why studies often report non-significant relationships between BPAQ and aggression overall), but that this is *not a problem* if you can identify the trials of interest (as we have). To be clear, our view is previous studies using the CRTT without selective analysis of the data have largely been examining non-aggressive behaviour – but that is not the case with the approach presented in our study. We recognize however that we did not communicate these points clearly enough in our original manuscript. Consequently, to more clearly address this issue we emphasize this issue in the introduction with specific reference to the merits of affect coding-informed trial rejection, which states:

“As well as using trial-to-trial changes in blast level as an empirical measure of aggressive behaviour, all participants completed a self-report measure of trait aggression (BPAQ-SF: the Buss and Perry Aggression Questionnaire, Short Form). In addition, we used OpenFace 2.0³³ (a machine learning program for automated Facial Action Coding³⁴) to identify prototypic displays of emotion during the blast initiation

response time window (blast initiation \pm 1 second; see Figure 1b), which allows us to assess the impact of each participant's emotional experience on behaviour (see Figure 1c). The differences between positive and negative emotions expressed during the experiment are visible in composite images (created with PsychoMorph³⁵; see Figure 1d). Importantly, because we measured facial expressions continuously throughout the task, we were able to examine how behaviour changed as a function of the emotions experienced by each partner. Further, and whenever possible, we use trial-level data in our analysis to maximize power. The use of trial level data enables us to examine the rich relationship between each partners' blast levels and affect over time, and potentially solve a historic problem in the field (i.e., how to study aggression in lab settings when the level of aggressive behaviour that can be ethically elicited is minimal). A solution here is possible because trial level data enables us to extract instances of aggression within a context of overall non-aggressiveness (i.e., an ethically and administratively palatable lab-based aggression paradigm)." (Page 6-7)

In addition, we have explicitly noted this issue in the results which states:

"As in previous studies^{e.g., 50} the BPAQ-SF (*max score*= 60^{cf. 36}) shows that a convenience sample of mostly undergraduate students scores lower ($m = 13.86$, $sd = 9.97$) on trait aggression than the general population. Similarly, as in previous research using the CRTT^{e.g., 51,52}, the standard experimental measure of aggression (all blast levels, averaged across trials for each participant) did not correlate with participants' self-reported trait aggression (see Figure 3a: $t(105) = 1.05$, $p = 0.29$, $r = 0.10$, $95\% CI = -0.09 - 0.29$, $bf = 0.40$). Poor correlations between the BPAQ and standard versions of the CRTT have been widely reported and discussed in the field^{37,38,53} and we will return to this in the Discussion.

However, our facial affect data (see Figure 3b) potentially explains the inconsistent CRTT findings in the present literature: prototypic displays of positive affect (happiness^{e.g., 28,29,54,55}) were far more frequent and intense than prototypic displays of negative affect (anger and disgust, both associated with conflict behaviour^{e.g., 28,29,54,55}; *mean positive affect* = 0.83, $SD = 0.77$; *mean negative affect* = 0.12, $SD = 0.21$; difference: $w = 8e+6$, $p > 2e-16$, $r = 0.50$, $95\% CI = 0.61 - 0.68$, $bf = 7e+484$). In other words, participants appear to find the CRTT task to be more fun than upsetting, a pattern also reflected in debrief feedback from participants. High negative emotion was present sufficiently often, however, (16% of trials) that its impact on aggression could be examined using selective averaging procedures, as shown below." (Page 15)

And the discussion section:

"Rather than examining aggressive *behaviour* in isolation, we recorded participants' facial expressions, which allowed behaviour to be examined as a function of trial-to-trial changes in emotional arousal, illuminating the dynamics of dyadic affective and behavioural processes during conflict. This allows us to identify and examine both essential components of reactive aggression, behavioural aggression and negative emotional arousal, rather than assuming reactive aggression from behaviour alone. To our initial surprise, assessment of the emotional expressions elicited during performance of the task highlighted that participants were most often experiencing positive emotion (i.e., happiness) suggesting that they enjoyed their participation. Nonetheless, by combining high time-resolution affect coding and selective averaging procedures, we were able to show that high levels of negative affect (indexed by prototypic displays of anger and disgust^{28,29,54,55}) predict increased levels of

aggression, regardless of which member of the couple was expressing the negative emotion.” (Page 26)

We hope that it is now clear that a) our paradigm does involve provocation by design, b) our paradigm does elicit aggression in practice, and indeed that there are no lower levels of aggression than in other comparable studies, and c) our analytic approach allows us to selectively examine the trials which contain aggression when it occurs.

4. One of the clearest findings from the current study is that delaying responding was associated with less aggressive behaviors. This finding replicates prior research (e.g., Finkel et al., 2009).

We thank the reviewer for highlighting this reference – we agree that it is related to our current manuscript and so we have now made reference to the overlap in findings in the discussion, the new text states:

“Most importantly, perhaps, we also showed that the introduction of a short delay (a forced break) between provocation and the opportunity for aggression led to a decrease in aggression. This finding builds on previous research⁶⁶, highlighting the possibility of developing practical interventions aimed at reducing intimate partner aggression.” (Page 24)

5. When discussing their delayed responding analyses, the authors imply they are testing competing hypotheses about delayed responding and rumination. I am not sold on the idea that the current study says much about the relationship between rumination and aggression. With rumination, it is not the mere passage of time, but it is the active processing and maintenance of provocation-related thoughts (e.g., Bushman et al., 2005; document (psu.edu)). For the results to be relevant to rumination, you’d actually need to know what is happening when people are delaying their responses. Are they simply stepping away and cooling down (not rumination)? Are they distracting themselves (also not rumination)? Or are they stewing about an instigation (rumination)? To test hypotheses about rumination, therefore, the authors would actually need to measure whether people are ruminating or instruct some people to ruminate (and instruct others not to ruminate) or something. You cannot simply equate delayed responding with rumination because those are not the same thing.

Although we had set up our study in terms of the tension between competing hypotheses, we agree with the reviewer’s critique of this framing, and have removed all reference to rumination within the context of our study, instead focusing on I³ as our source of a hypothesis (as highlighted above in response to reviewer 1, point 2). Whilst we agree with the reviewer that it would be interesting to carry out additional studies focused specifically on the manipulations of rumination (e.g., instructing some people to ruminate, others not to) this issue goes beyond the aims of our original study.

As we hope is now clear, our key aim was to examine whether the introduction of a forced break led to a reduction in aggression – and we did explicitly manipulate the presence of a forced break for this reason. We hope that the addition of the follow up study (which replicates our key findings) serves to reassure the reviewer that this manipulation does indeed have a strong and reproducible effect in reducing aggression. We also take the reviewers comments regarding the question of mechanism to heart and have added a number of additional analyses that speak to this issue (as noted in response to reviewer 1, points 3 and 7; response to reviewer 2, point 1).

6. I am not sure if there is a clear enough take-home message for this to be a stand-alone study though. To make stronger claims about intimate partner violence, I feel like there needs to be better conditions for aggression to occur (e.g., an instigation).

As noted above (response to reviewer 1, points 2 and 3), we believe that our paradigm does involve instigation and so we have now made this clearer in the introduction (and throughout). In addition, at the request of the Editor we have been careful in distinguishing between intimate partner violence and intimate partner aggression (where we only discuss our study in terms of the latter).

More broadly, we note the ongoing issue here, common to all aggression research: researchers need to provoke aggression (or at least a reasonable proxy) in the lab – but only within the ethical frameworks within which our work is approved.

We strongly believe that our development of the CRTT makes it considerably more ecologically valid than in its standard form. Specifically, we have real couples, standing in a room together, looking at one another face-to-face, sending real (very loud) sound blasts to each other's headphones, where they can see the response that these sound blasts invoke. In this context, we do not believe it would be reasonable (nor would we likely get ethical permission) to deliberately incite conflict between our couples before presenting them with an opportunity to inflict pain on each other.

As noted above (in response to reviewer 1, points 1, 2, and 3), however, we understand the criticism of the current study as a stand-alone experiment, and hope that our addition of replication data provides reassurance about the strength and rigour of our findings. In addition, we hope that the new qualitative data makes clear that our participants experienced anger (sometimes), as well as experiencing the louder blasts as unpleasant and aggressive.

7. And for there to be stronger inferences about why delayed responding works, I feel like there would need to be methods that are designed to assess why delayed responding leads to less aggression.

We agree that any article aiming to make inferences about findings requires a clear proposed mechanism supported by data. Our original manuscript did include data pertaining to a proposed mechanism, but we did not adequately highlight this data and make our proposed mechanism explicit. To be clear, we propose that forced breaks reduce aggression by first preventing impulsive action in a provoked state and then reducing negative affect before allowing a behavioural response. To make our proposal, and the data supporting it, clear, we have made a series of substantial changes:

- a) We have laid out the theoretical groundwork for this interpretation in the intro:

“Critically, emotional arousal and impulsivity are linked, such that acute negative affect (e.g., an angry state) is associated with failures of self-control. Within individuals, as a personality trait, the tendency to act rashly when upset is called ‘negative urgency’^{e.g., 4,5,17,18,20}, and it has been implicated as a central contributor to intimate partner aggression.” (Page 3),
- b) We have highlighted the reduction of emotion during the break in Figure 4, Panel e by simplifying our reporting of these data,
- c) We have included (multi-level) Actor-Partner Interdependence Modelling (following the recommendation of Reviewer 2, point 5) which illuminates the relationship between each partner's affect over time in the immediate response (Figure 7a) and the forced break (7c) conditions. Both models are also presented with regressions showing the impact (or lack of impact) of both partner's negative affect on blast level (panels b and d),
- d) We have included a mediation analysis to assess the impact of the forced break condition on blast level selection directly and as mediated by negativity. Our results

suggest a complete mediation of forced breaks by negativity, providing further evidence that forced breaks reduce aggression by reducing negativity (or more precisely, by preventing action until negativity has had time to reduce),

- e) We have added a section to the results dedicated to discussing points 7a-d (the additions are too long to quote here in full, but for example):
- “The above analysis has demonstrated that forced breaks reduce negative affect and blast levels (behavioural aggression) under conditions of high negative affect; however, the potential mechanism behind forced breaks can be further illuminated by statistical modelling of the data. Specifically, we employed multi-level, longitudinal Actor-Partner Interdependence Modelling^{e.g., 56,57} (APIM) to examine the interplay of both partner’s emotions over time. APIM allows us to assess the impact of the previous winner’s and loser’s negative affect (at R_{i-1}) on the current winner’s and loser’s negative affect (at R_i), remembering that each player oscillates between winning and losing over time (as described above). Specific model parameters and full statistics for all following models are provided in Table 3.” (Page 20),
- f) We have conducted a strict replication analysis (using all new data from a follow-on study) of the main findings with a focus on those findings most important to our proposed mechanism, shown in a new Figure 9 and presented in the results (again, in brief):
- “Having established a link between the expression of negative emotion and aggressive behaviour using our face-to-face Competitive Reaction Time Task we carried out a follow-on study (as outlined in the methods) that allowed us to ask whether the effects were replicable. As in the primary study, results of the replication provide strong evidence for the alternative hypothesis^{60,61} (Table 5).” (Page 22),
- g) We have provided additional qualitative data from our follow-on (replication) study that speak to the participant’s perceptions of the impact of the forced breaks on their thoughts, emotions, and behaviour (they also speak to the ecological validity of the task; presented in Table 6 and mentioned in the discussion):
- “Whilst our noxious sound is quite unlike acts of physical aggression seen in real-world settings, participants reported it as aversive during debriefing, even expressing fear of receiving blasts and guilt after sending them (see Table 6). This suggests that delivering sound blasts is a reasonable proxy for aggressive behaviour.” (Page 24),
- h) Finally, we have made several modifications to the discussion that flush out our proposed mechanism and highlight how our methodological modifications support the investigation of the mechanism, such as:
- “Previous studies have approached the present design, bringing couples into a lab to play a fake version of the CRTT in separate rooms (usually they are told that their blasts/shock selections will be delivered to their partner, and vice versa, but they are not)^{e.g., 18,68} or bringing strangers into a lab and having them really send each other blasts, but from separate rooms⁸. By examining couples together, face-to-face and in-person, the present study offers a significant methodological improvement, increasing the realism of the standard CRTT while maintaining its experimental control. Although we employed a modest sample size (53 couples in the primary study and 29 in the replication) our approach also allowed us to maximise power by focusing on trial level data (which gave us thousands of observations for most analyses). Critically, we also employed an innovative and underused approach to the analysis of behaviour.”

[...] “This allows us to identify and examine both essential components of reactive aggression, behavioural aggression and negative emotional arousal, rather than assuming reactive aggression from behaviour alone.” (Page 26),

“Based on the current findings we propose that forced breaks reduce aggression by first blocking the expression of impulsive behavioural aggression while participants are in a provoked state. Then, because negative affect reduces over time, when a response is expressed, it will tend to be less aggressive (assuming no further provocation has occurred). We expect, of course, that a range of other factors, including individual traits and situational variables (such as the couple’s escalation/de-escalation asymmetry), will also shape the outcomes of couples’ conflict. Nonetheless, in practice, our results provide strong evidence for the role of impulsivity and negative affect in intimate partner aggression, at least during the kind of short-term, acute conflict present in our experiment.” (Page 28)

We believe these changes have substantially improved our manuscript. As mentioned above, and despite these improvements, we have removed any inferences about intimate partner violence based on this proposed mechanism as it is preliminary, and no lab-based aggression task can really examine violence, as requested by the Editor.

Comments from Reviewer 2:

- 1. I had some questions about the theoretical justification for the study. The authors contrast negative urgency and rumination as factors involved in couples' conflict/aggression. Specifically, they say that a break would reduce the effect of negative urgency but accentuate the effect of rumination. I know that previous research has clearly distinguished trait and state rumination (e.g., Nolen-Hoeksema et al., 2008), so I think rumination could make sense here (although the idea of ruminating for 5-15 seconds may not make sense), but I think of negative urgency as a trait. That is people high in negative urgency behave impulsively when they are experiencing a state of negative emotion.**

We appreciate your concerns about our theoretical justifications and agree that they are not adequate. We especially agree that rumination does not provide a compelling theoretical explanation of our findings. As such, we have removed all mention of it (as mentioned in, and requested by, Reviewer 1, Point 5). Regarding negative urgency, we agree that we did not adequately explain/define our conception of negative urgency in our original manuscript, and thus our discussion of it was no-doubt confusing. To be clear, we are specifically interested in (and examine) state-level impulsivity during a provoked state, which we manipulate by blocking impulsive action (with a forced break). To address your concern (and in line with comments from reviewer 1), and to clarify the focus (and limits) of our study, we have made substantial changes to the framing in the introduction and discussion:

“Intimate Partner Aggression is widespread, with around a third of dating couples at universities engaging in some form of physical aggression during conflict⁹. As a result, researchers and practitioners are increasingly focused on understanding the social and emotional interactions that occur between partners during everyday conflict, and how these predict the escalation towards aggression¹⁰. Aggression between partners is frequently reactive in nature – that is, it is impulsive^{e.g., 11,12}, and is characterised by strong negative emotional arousal^{4,5,13}. Critically, emotional arousal and impulsivity are linked, such that acute negative affect (e.g., an angry state) is

associated with failures of self-control. As a personality trait, the tendency to act rashly when upset is called ‘negative urgency’^{e.g., 4,5,11,13,14}, and it has been implicated as a central contributor to intimate partner aggression. By contrast, the systematic pattern of abuse sometimes known as ‘intimate terrorism’ or ‘coercive controlling violence’ is less strongly associated with impulsivity and we therefore use the term ‘intimate partner aggression’, rather than ‘intimate partner violence’, to describe our research focus¹⁵.” (Page 3)

“Based on the current findings we propose that forced breaks reduce aggression by first blocking the expression of impulsive behavioural aggression while participants are in a provoked state. Then, because negative affect reduces over time, when a response is expressed, it will tend to be less aggressive (assuming no further provocation has occurred). We expect, of course, that a range of other factors, including individual traits and situational variables (such as the couple’s escalation/de-escalation asymmetry), will also shape the outcomes of couples’ conflict. Nonetheless, in practice, our results provide strong evidence for the role of impulsivity and negative affect in intimate partner aggression, at least during the kind of short-term, acute conflict present in our experiment.” (Page 28)

- 2. Along these lines, a recent study found that negative urgency was positively related to aggression on the Taylor Aggression Task when participants were displaying greater negative affect (as assessed with FACS) but not when they were displaying less negative affect. Based on this, I’m not sure that I understand the authors’ argument that negative urgency would take charge with no break. I would see it as affect would be driving that state affect but perhaps for people high in negative urgency.**

We agree that negative affect is driving aggression in the immediate response condition, and we failed to clarify our interpretation of the role of impulsive action during a provoked state. We have made substantial changes to our manuscript to clarify these points, as described above in Reviewer 2, Point 1. We have also added substantial analysis and text to our manuscript with the goal of explaining how we believe impulsivity during a state of provocation contributes to aggression, and how we have mitigated it in our forced break condition (described in Reviewer 1, Point 7).

- 3. I also found it slightly confusing that you talk about negative urgency and rumination, but then they aren’t measured or manipulated in the study. I guess these suppositions would better fit in the discussion given the study design.**

We appreciate this confusion and can see that we did not adequately motivate/theoretically ground our methods in the introduction. As mentioned in Reviewer 2, Points 1 and 2, we have taken great pains to correct this weakness in our manuscript (Reviewer 1, Point 7; Reviewer 2, Points 1 and 2).

Further, we argue that we manipulated state impulsivity amidst a provoked state by experimentally blocking behavioural aggression during acute instances of impulsivity in the forced break condition. We have made substantial changes to the manuscript to clarify this point, included new analysis and data pertaining to our proposed mechanism (as described in Reviewer 1, Point 7) and evident throughout our updated manuscript (for example, here again in the discussion):

“Based on the current findings we propose that forced breaks reduce aggression by first blocking the expression of impulsive behavioural aggression while participants

are in a provoked state. Then, because negative affect reduces over time, when a response is expressed, it will tend to be less aggressive (assuming no further provocation has occurred). We expect, of course, that a range of other factors, including individual traits and situational variables (such as the couple's escalation/de-escalation asymmetry), will also shape the outcomes of couples' conflict. Nonetheless, in practice, our results provide strong evidence for the role of impulsivity and negative affect in intimate partner aggression, at least during the kind of short-term, acute conflict present in our experiment." (Page 28)

Finally, we agree that we did not measure trait negative urgency, which we now acknowledge in our limitations section:

"In addition, it could also be valuable to include a trait measure of negative urgency in future studies on forced breaks, as variability in personality between participants might moderate the effect of forced breaks on negative affect and aggression." (Page 29)

4. I had a few comments on the methods. I would like to see the internal consistency of the measures. It's important to show that the measures are reliable.

We agree that these are important metrics to include. We have now calculated and included a description of the internal consistency for all of our measures:

"Cronbach's Alpha^{44,45} indicated that internal consistency was "robust" or better⁴⁶ for each measure (FTF-CRTT, 30 items, $\alpha = .925$; negativity, 30 items, $\alpha = .866$; BPAQ-SF, 12 items, $\alpha = .835$)" (Page 14)

5. From what I can tell, the authors did not take into account that the data came from couples who were face-to-face. I would expect some kind of multi-level modeling to account for the covariation in the data. I would also think that actor-partner modeling would be of interest because this would allow you to see how one partner's affect display is correlated with the other partner.

We thank the reviewer for suggesting these additional analyses. We had conducted MLM analysis but did not include it in our initial submission because of the (soft) page limit. The Editor has allowed us additional space, so we have now included a L-APIM analysis in the main text, along with a mediation analysis (and Figures 7 and 8, Tables 3 and 4); the full addition is too long to quote here, but for example:

"Specifically, we employed multi-level, longitudinal Actor-Partner Interdependence Modelling^{e.g., 56,57} (APIM) to examine the interplay of both partner's emotions over time. APIM allows us to assess the impact of the previous winner's and loser's negative affect (at R_{i-1}) on the current winner's and loser's negative affect (at R_i)." (Page 20),

"Finally, to further illustrate the potential mechanism of forced breaks, we conducted a simple mediation analysis^{e.g., 58,59} to examine the direct and indirect impact of the forced break condition on blast level selection (behavioural aggression) through winner negativity." (Page 21)

We hope these additional analyses address your concern; we believe they have substantially improved our manuscript.

6. It seems that the authors did not include a zero or no noise blast. This makes it more difficult to interpret noise blast selection as aggression since there is no non-aggressive response. This limitation should be noted.

We agree that not having a zero-noise blast is a potential limitation of the current study. As per recommendation by Elson et. al. (2014), we did not include a zero-blast condition in order to avoid skewing mean-based aggression scores. Nevertheless, to use or not to use a zero-blast option remains an important consideration, and so we have mentioned this in our limitations section:

“A common critique of the CRTT is that there is no non-aggressive response option. Participants must give a sound blast, which makes it difficult to infer aggressive intent from the use of a sound blast. We chose our response format based on recommendations from previous work³⁵, and while participants could not choose ‘no response’, in practice they did use the full range of blasts available to choose how much discomfort to inflict. Nonetheless it is important to consider the fact that when inflicting discomfort is facilitated (by design) by an experimental setup, it limits the extent to which we can make claims about ‘real-world’ aggression.” (Page 29)

7. It may be because the method section is later in the document, but I felt like I needed a data analysis section to link the analyses to the research questions. The results started with several analyses that I was not expecting based on the introduction and it decreased the readability.

Our methods appeared later in the document because the last version of the manuscript came via an automatic transfer from another journal with different formatting requirements. We have now updated the manuscript to fit the expected format (with methods in the main text, where they belong). Regarding readability, due to our substantially revised theoretical framing and analysis, we cannot address this concern in its original form. However, we appreciate this feedback and have taken it seriously. We believe that our introduction now flows more intuitively into our methods, results, and discussion section; we hope that you agree.

8. I think the authors did not include/review some relevant studies. The authors are correct that this is the first face-to-face couples aggression study, but there are several studies that have used aggression paradigms to look at couples' aggression. For example, Eckhardt et al. (2021) found that alcohol-induced aggression ostensibly against a partner was mediated by cognition, not affective expression. As noted above, Bresin et al., (2022) examined negative urgency and negative affect facial expression in laboratory aggression ostensibly against a partner. Watkins et al (2015) manipulated rumination versus reappraisal in a couples version of the Taylor aggression paradigm. The current study adds to this work, but this work seems relevant in interpreting the current results.

We thank the reviewer for highlighting these additional relevant studies. We have now cited and discussed them in the manuscript, where appropriate:

“Previous studies have approached the present design, bringing couples into a lab to play a fake version of the CRTT in separate rooms (usually they are told that their blasts/shock selections will be delivered to their partner, and vice versa, but they are not)^{e.g., 18,68} or bringing strangers into a lab and having them really send each other blasts, but from separate rooms⁸.” [...] “This allows us to identify and examine both essential components of reactive aggression, behavioural aggression and negative emotional arousal, rather than assuming reactive aggression from behaviour alone (a

method used successfully in previous studies with different focuses^{e.g., 28,76}.” (Page 26)

“Fundamentally, therefore, we are optimistic about the implications of the current findings. Experimental research involving couples can reveal the complex dyadic processes influencing intimate partner aggression^{28,68,76,82}, adding to growing awareness of the importance of negative emotional arousal and emotional co-regulation as key drivers of aggression, knowledge that should ultimately contribute to the development of effective interventions. “ (Page 30)

11th Jul 24

Dear Mx McCurry,

Your manuscript titled "Both Partners' Negative Emotion Drives Aggression During Couples' Conflict" has now been seen by our reviewers, whose comments appear below. In light of their advice I am delighted to say that we are happy, in principle, to publish a suitably revised version in *Communications Psychology* under the open access CC BY license (Creative Commons Attribution v4.0 International License).

We therefore invite you to revise your paper one last time to address the remaining concerns of our reviewers and a list of editorial requests. At the same time we ask that you edit your manuscript to comply with our format requirements and to maximise the accessibility and therefore the impact of your work.

EDITORIAL REQUESTS:

SUBMISSION INFORMATION:

OPEN ACCESS:

Communications Psychology is a fully open access journal. Articles are made freely accessible on publication under a CC BY license (Creative Commons Attribution 4.0 International License). This

license allows maximum dissemination and re-use of open access materials and is preferred by many research funding bodies.

For further information about article processing charges, open access funding, and advice and support from Nature Research, please visit <https://www.nature.com/commspsychol/article-processing-charges>

At acceptance, you will be provided with instructions for completing this CC BY license on behalf of all authors. This grants us the necessary permissions to publish your paper. Additionally, you will be asked to declare that all required third party permissions have been obtained, and to provide billing information in order to pay the article-processing charge (APC).

* **DATA AVAILABILITY:**

[link redacted]

Best regards,

Marike

Marike Schiffer, PhD

Chief Editor

Communications Psychology

REVIEWERS' COMMENTS:

Reviewer #1 (Remarks to the Author):

I have now had a chance to review the revised manuscript “Both partners’ negative emotion drives aggression during couples’ conflict” for possible publication in Communications Psychology. This version of the manuscript is greatly improved. I also wanted to point out that the response letter was thorough and well-written. And the additional study also was a nice addition.

I have a generally positive evaluation of the manuscript, which is something that, upon reading my previous review, I wish had been more explicit. Nonetheless, I do have a few more questions/comments that I would like the authors to consider.

Were partners able to see each other’s’ sound blast selections? During the CRTT, sometimes participants are told what blast level their partner chose. Or, in the case of the face-to-face nature

of the current study, I could imagine partners peeking at one another's keyboards to see which button their partner pressed (or even talking to one another about their blast level selection). If so, I wonder if this would lead participants to "match" one another's selections more than what might occur in a more naturalistic setting. This is not a critique per se, but this is a detail that ought to be included in the Procedures section.

With the addition of the replication study, can the authors please confirm that all the conditions, measures, and exclusions are reported (similarly to the 21-word statement)? It is fine if the authors were able to "use" an unpublished study to supplement their primary study, but I want to ensure that they are providing the reader with all the necessary information. To be clear, I don't suspect the authors are hiding anything, but explicitness would be helpful for readers to be able to gauge how well the replication study corroborates their primary study.

A nuance that could be added is that the current studies speak to impulsive aggressive behaviors. There are several aggressive behaviors that actually require self-control or for partners to be more playful. For example, secretly poisoning your partner requires a great deal of self-control. Or, as a less extreme example, gas lighting your partner might be a form of psychological/emotional aggression. At some point, the authors may want to introduce a nuance that delayed responding may not lead to a decrease in aggression (broadly speaking), but delayed responding may lead to a decrease in impulsive aggression. This may not need to be a separate point, but may be woven into the Intro or Discussion.

I do have a few thoughts about style. Although I can appreciate the skill that goes into making these figures, there are several figures that seem unnecessary or may best be placed in online supplements. It seemed like a lot of figures for this manuscript and I found myself spending a lot of time trying to make sense of some of the figures. For example, Figure 6 has several panels that show the data from a few couples. Although interesting, these don't add much for me personally and could be removed without (IMO) losing anything.. I would select a handful of figures that best highlight the main contributions of the studies to include in the main body of the manuscript. With that said, I am not an author and I would defer to the actual authors to make these decisions.

As always, I ran this manuscript through statcheck. There were two statistics that were flagged as "inconsistent." The t-statistic " $t(326) = 4.1, p > 0.0001$ " and the t-statistic " $t(345) = 6.57, p = 1.8e-10$ ". Please have the authors double-check these to make sure they are correct.

Reviewer #2 (Remarks to the Author):

The authors have addressed my concerns. I do not have any remaining comments.

Comments from Reviewer 1:

- 1. I have now had a chance to review the revised manuscript “Both partners’ negative emotion drives aggression during couples’ conflict” for possible publication in Communications Psychology. This version of the manuscript is greatly improved. I also wanted to point out that the response letter was thorough and well-written. And the additional study also was a nice addition. I have a generally positive evaluation of the manuscript, which is something that, upon reading my previous review, I wish had been more explicit.**

We thank you for your valuable comments from the previous round of review; we agree that your feedback (and the feedback from R2) substantially improved our manuscript.

- 2. Were partners able to see each other’s’ sound blast selections? During the CRTT, sometimes participants are told what blast level their partner chose. Or, in the case of the face-to-face nature of the current study, I could imagine partners peeking at one another’s keyboards to see which button their partner pressed (or even talking to one another about their blast level selection). If so, I wonder if this would lead participants to “match” one another’s selections more than what might occur in a more naturalistic setting. This is not a critique per se, but this is a detail that ought to be included in the Procedures section.**

We agree that this topic is worth explicating in the procedures section. We did not show partners’ blast selections to each other, but we also did not attempt to prevent them from seeing them or talking about it. We agree that this likely increased the accuracy of participants’ guesses of what their partner had chosen (compared to trying to guess from the volume alone). From our perspective, this is not an issue of ecological validity, because selecting discrete aggression levels is the least ecologically valid aspect of the CRTT to begin with. However, within the context of the CRTT, being able to see or ask what specific blast level someone chose would certainly effect retaliatory precision from a statistical perspective. We have updated our methods section with the following to illuminate this topic:

“We did not show participants the blast levels chosen by their partner, but we also did not prevent them from peeking at their partner’s keyboard or talking to each other about the discrete numbers selected. Because the loser received whichever blast level the winner chose, we had no reason to try to hide this from the loser, and no practical way to prevent participants from discussing it. In fact, we chose to use non-noise isolating headphones specifically so that partners could talk to each other during the game, to increase ecological validity, and so that the winner could hear (from sound leaked from the loser’s headphones) that they were really in control of the blast level.” Pg. 11

- 3. With the addition of the replication study, can the authors please confirm that all the conditions, measures, and exclusions are reported (similarly to the 21-word statement)? It is fine if the authors were able to “use” an unpublished study to supplement their primary study, but I want to ensure that they are providing the reader with all the necessary information. To be clear, I don’t suspect the authors are**

hiding anything, but explicitness would be helpful for readers to be able to gauge how well the replication study corroborates their primary study.

We are pleased to confirm that we reported all conditions, measures, and exclusions for both the primary and the replication study in our manuscript and in our reporting summary. We are not sure what “21-word statement” you are referring to, but we have added the following statement to the methods sections in order to assure the reader that we have provided all information about the follow-on study:

“Here, for the sake of brevity, we only include data from the follow-on study that provides a replication of the original experiment. For the replication data we report all measures and experimental conditions in the present article and the associated reporting summary.” Pg. 13

- 4. A nuance that could be added is that the current studies speak to impulsive aggressive behaviors. There are several aggressive behaviors that actually require self-control or for partners to be more planful. For example, secretly poisoning your partner requires a great deal of self-control. Or, as a less extreme example, gas lighting your partner might be a form of psychological/emotional aggression. At some point, the authors may want to introduce a nuance that delayed responding may not lead to a decrease in aggression (broadly speaking), but delayed responding may lead to a decrease in impulsive aggression. This may not need to be a separate point, but may be woven into the Intro or Discussion.**

We agree with your interpretation that our study is more concerned with impulsive forms of aggression than premeditated/proactive aggression. We argue, however, that we have made this point explicitly clear throughout the document. For example:

“Aggression between partners is frequently reactive in nature – that is, it is impulsive^{e.g., 11,12}, and is characterised by strong negative emotional arousal^{4,5,13}. Critically, emotional arousal and impulsivity are linked, such that acute negative affect (e.g., an angry state) is associated with failures of self-control. As a personality trait, the tendency to act rashly when upset is called ‘negative urgency’^{e.g., 4,5,11,13,14}, and it has been implicated as a central contributor to intimate partner aggression. By contrast, the systematic pattern of abuse sometimes known as ‘intimate terrorism’ or ‘coercive controlling violence’ is less strongly associated with impulsivity and we therefore use the term ‘intimate partner aggression’, rather than ‘intimate partner violence’, to describe our research focus¹⁵.” Pg. 3

“Consequently, if reactive aggression between couples requires negative arousal (influenced by both partners) and impulsivity, we reasoned that it should be possible to reduce behavioural aggression by experimentally blocking impulsive action while participants are in a provoked state. In principle, impulsive action can be prevented by introducing a ‘forced break’ period (e.g., a brief experimentally imposed delay) between provocation and the opportunity for aggression. If impulsivity during a provoked state is critical, we would expect a forced break to limit aggression by blocking impulsive action and reducing negative emotion. In essence, assessing the

consequences of an experimentally imposed forced break on the dynamics of dyadic conflict provides a strong test of the influence of both negative affect and impulsivity during provocation, allowing us to examine current accounts of Intimate Partner Aggression.” Pg. 3-4

“In response, the current study asks three questions relating to the role of dyadic affective and behavioural processes in reactive aggression between romantic partners: First, does the introduction of a forced break (i.e., a brief experimentally manipulated delay ranging from five to 15 seconds) between provocation and the opportunity for aggression prevent impulsive action and reduce negative emotions, thereby decreasing aggression?” Pg. 5

“Within the theoretical aggression literature^{e.g., 6,7}, it has been posited that both inter- and intra- personal affective processes, as well as impulsivity^{11,14}, impact aggression. Based on the current findings we propose that forced breaks reduce aggression by first blocking the expression of impulsive behavioural aggression while participants are in a provoked state. Then, because negative affect reduces over time, when a response is expressed, it will tend to be less aggressive (assuming no further provocation has occurred).” Pg. 28

“Nonetheless, in practice, our results provide strong evidence for the role of impulsivity and negative affect in intimate partner aggression, at least during the kind of short-term, acute conflict present in our experiment.” Pg. 28

“Specifically, forced breaks can only be effective in reducing reactive aggression. Forced breaks should not be expected to have any influence on the systematic, premeditated forms of abuse used to exert long-term control over a partner.” Pg. 29

Because we have already been quite clear about the role of impulsivity in the current studies, and the limitations of the generalizability of the current findings based on that fact, we have not amended our manuscript in response to this concern.

5. I do have a few thoughts about style. Although I can appreciate the skill that goes into making these figures, there are several figures that seem unnecessary or may best be placed in online supplements. It seemed like a lot of figures for this manuscript and I found myself spending a lot of time trying to make sense of some of the figures. For example, Figure 6 has several panels that show the data from a few couples. Although interesting, these don't add much for me personally and could be removed without (IMO) losing anything.. I would select a handful of figures that best highlight the main contributions of the studies to include in the main body of the manuscript. With that said, I am not an author and I would defer to the actual authors to make these decisions.

We agree that we included a lot of figures, and we appreciate your concern that some are perhaps superfluous. However, we would argue that these inclusions are reasonable given the amount of analysis and complexity of the data. Specifically, the Figure 6 plots showing

individual couples illustrate a fascinating pattern in the data that would be extremely difficult to convey with words alone (here, the image is worth 1000 words). Finally, Communications Psychology requests that all figures remain in the main text and any irrelevant figures be wholly excluded (rather than placed in the supplemental information). For these reasons, we will leave all figures in the manuscript at this point. We appreciate this feedback none the less, and we took this decision seriously.

6. As always, I ran this manuscript through statcheck. There were two statistics that were flagged as “inconsistent.” The t-statistic “ $t(326) = 4.1, p > 0.0001$ ” and the t-statistic “ $t(345) = 6.57, p = 1.8e-10$ ”. Please have the authors double-check these to make sure they are correct.

We appreciate your care and rigour when reviewing our manuscript and we are very grateful that you noticed these small errors. In the case of “ $t(326) = 4.1, p > 0.0001$ ”, there are two minor issues. First, $t = 4.6$, not 4.1; and the p-value is not $>$ (more than)0.0001, it is $<$ (less than)0.0001. With these corrections, the statistic now read “ $t(326) = 4.6, p < 0.0001$ ” and is no longer flagged as inconsistent by statcheck.

The second instance (“ $t(345) = 6.57, p = 1.8e-10$ ”), however, is correct. We have triple checked our analysis and recording of the statistic and we have reported them as they present in r. Further, we manually calculated a p-value given the t-statistic and degrees of freedom, and this manual calculation is consistent with our results. Our best guess for the flag by statcheck is that we report exact p-values when possible, even when they are well below 0.0001, which is perhaps violating the expectations of the program. Indeed, when we upload our manuscript with “ $p < 0.0001$ ” instead of “ $p = 1.8e-10$ ”, it is no-longer flagged as inconsistent. We have chosen to retain “ $p = 1.8e-10$ ” in our manuscript as we are satisfied that the provided stats are accurate and statcheck is reacting to an unusually precise p-value rather than an error in reporting.

Comments from Reviewer 2:

1. The authors have addressed my concerns. I do not have any remaining comments.

Thank you for the valuable feedback you provided in the last round of review. We’re glad our revisions satisfied your concerns.